# Signal transducer and activator of transcription 5 (STAT5) paralog dose governs T cell effector and regulatory functions

Alejandro Villarino[1]*, Arian Laurence[1], Gertraud W Robinson[2], Michael Bonelli[1], Barbara Dema[1], Behdad Afzali[1], Han-Yu Shih[1], Hong-Wei Sun[1], Stephen R Brooks[1], Lothar Hennighausen[2], Yuka Kanno[1], John J O'Shea[1]

[1]Molecular Immunology and Inflammation Branch, National Institute of Arthritis, Musculoskeletal and Skin Diseases, National Institutes of Health, Bethesda, United States; [2]Laboratory of Genetics and Physiology, National Institute of Diabetes, Digestive and Kidney Diseases, National Institutes of Health, Bethesda, United States

**Abstract** The transcription factor STAT5 is fundamental to the mammalian immune system. However, the relationship between its two paralogs, STAT5A and STAT5B, and the extent to which they are functionally distinct, remain uncertain. Using mouse models of paralog deficiency, we demonstrate that they are not equivalent for CD4[+] 'helper' T cells, the principal orchestrators of adaptive immunity. Instead, we find that STAT5B is dominant for both effector and regulatory (Treg) responses and, therefore, uniquely necessary for immunological tolerance. Comparative analysis of genomic distribution and transcriptomic output confirm that STAT5B has fargreater impact but, surprisingly, the data point towards asymmetric expression (i.e. paralog dose), rather than distinct functional properties, as the key distinguishing feature. Thus, we propose a quantitative model of STAT5 paralog activity whereby relative abundance imposes functional specificity (or dominance) in the face of widespread structural homology.

*For correspondence: alejandro.villarino@nih.gov

## Introduction

Signal transducers and activators of transcription (STAT) family proteins are an evolutionarily conserved set of transcription factors, which operate downstream of cytokine and hormone receptors to convert extracellular stimuli into biochemical signals that instruct gene expression (*Villarino et al., 2015*; *Stark and Darnell, 2012*). In mammals, STAT5 is unique because it is encoded by two genes, termed *Stat5a* and *Stat5b*, derived from a relatively recent duplication event (*Wang and Levy, 2012*). In fact, the ancestral STAT5 gene appears to have duplicated on two separate occasions during vertebrate evolution. Once in teleosts, resulting in two paralogs on different chromosomes, and again in eutherians, resulting in the two contiguous mammalian paralogs (*Liongue et al., 2012*).

Because of their recent divergence, STAT5A and STAT5B are homologous at the DNA, RNA and protein levels, which has led to persisting questions about whether they are redundant or functionally distinct. Genetically engineered mice lacking *Stat5a* or *Stat5b* have provided compelling evidence for both arguments. On one hand, there are phenotypic differences; *Stat5a*-deficient mice exhibit poor mammary function (*Liu et al., 1997*), reduced hematopoietic stem cell proliferation (*Zhang et al., 2000*) and diminished antibody class switching (*Kagami et al., 2000*), while *Stat5b*-deficient mice exhibit dwarfism (*Udy et al., 1997*), more pronounced lymphopenia, and greater

**eLife digest** The immune system in mammals is one of the most complex networks in the animal kingdom. One way that its many components communicate is via proteins called cytokines, which are released by cells and detected by receptors on the surface of other cells. This leads to the activation of signals inside the responding cells that alter the activity of genes and, ultimately, direct how they behave.

STAT5 is a signal protein that is activated when certain cytokines bind to receptors on the cell surface. Consequently, it is an attractive target for drug therapies that seek to alter immune responses and there is keen interest in understanding how it works. It is an unusual protein in that there are two versions – termed STAT5A and STAT5B – that are produced by two separate genes. Together, STAT5A and STAT5B are fundamental to the immune system but there is considerable debate about whether they perform the same job or have distinct roles.

Villarino et al. used a combination of genetic and genomic approaches to investigate how both versions of STAT5 work in mice. The experiments show that STAT5B plays a much bigger role in immune cells than STAT5A. Unexpectedly, the experiments indicate that the disparity is not due to differences in protein activity, but is caused by differences in the amount of these proteins in cells.

Villarino et al.'s findings resolve longstanding questions about the relationship between STAT5A and STAT5B within the immune system. A logical next step is to find the molecular mechanisms responsible for causing different amounts of STAT5A and STAT5B to be produced in immune cells. Future work will also compare the roles of STAT5A and STAT5B in non-immune cells and explore whether it might be possible to develop therapies that specifically target one version and not the other.

---

defects in cytokine-driven lymphocyte proliferation (*Moriggl et al., 1999b*; *Imada et al., 1998*). On the other hand, deletion of *Stat5a* and *Stat5b* has comparable effects on some physiological processes, such as eosinophil recruitment (*Kagami et al., 2000*), and the most dramatic phenotypes, such as infertility, anemia and perinatal lethality, are evident only in mice lacking both paralogs, which implies redundancy and/or cooperativity (*Teglund et al., 1998*; *Socolovsky et al., 1999*; *Cui et al., 2004*). Genome-wide DNA-binding profiles also support both viewpoints. The target repertoires for STAT5A and STAT5B mostly overlap, which implies redundancy, but there are also a subset of sites that may be differentially bound, which implies specificity (*Liao et al., 2008*; *2011*; *Yamaji et al., 2013*; *Kanai et al., 2014*). Consistent with the latter point, humans with germline mutations in *STAT5B* exhibit a range of clinical abnormalities, indicating that *STAT5A* cannot compensate for some vital functions (*Kanai et al., 2012*).

Compound STAT5 deficiency manifests striking immunological abnormalities in mice, most notably lymphopenia, splenomegaly and autoimmunity. These are typically attributed to its role downstream of the common gamma chain (γc) receptor and its dedicated Janus kinase, Jak3 (*Moriggl et al., 1999b*; *Snow et al., 2003*; *Yao et al., 2006*). The γc is shared by 6 different cytokines, IL-2 IL-4, IL-7, IL-9, IL-15 and IL-21, each of which employs a unique co-receptor subunit that determines which cell types can respond (*Rochman et al., 2009*). γc cytokines impact all lymphocytes but have been most extensively studied in CD4[+] 'helper' T cells, the key orchestrators of adaptive immunity. Among the many functions ascribed to the γc-STAT5 axis in this lineage are the ability to promote Th1- and Th2-type effector responses, to support T cell memory, to promote activation-induced cell death, to suppress Th17-type and T follicular helper cell (Tfh) responses, and to promote T regulatory cell (Treg) responses (*Moriggl et al., 1999a*; *Liao et al., 2008*; *2011*; *Dooms et al., 2007*; *Zhu et al., 2003*; *Kagami et al., 2001*; *Lenardo, 1991*; *Laurence et al., 2007*; *Ballesteros-Tato et al., 2012*; *Johnston et al., 2012*; *Mahmud et al., 2013*).

To assess redundancy between STAT5 paralogs, we developed a mouse model where STAT5A and/or STAT5B were reduced but not absent, allowing us to compare their respective functions while avoiding the confounding lymphopenia associated with complete STAT5 deficiency. These studies reveal STAT5B as the dominant paralog in helper T cells; exhibiting far greater impact on pathogenic effector and host-protective regulatory responses and, therefore, uniquely required for

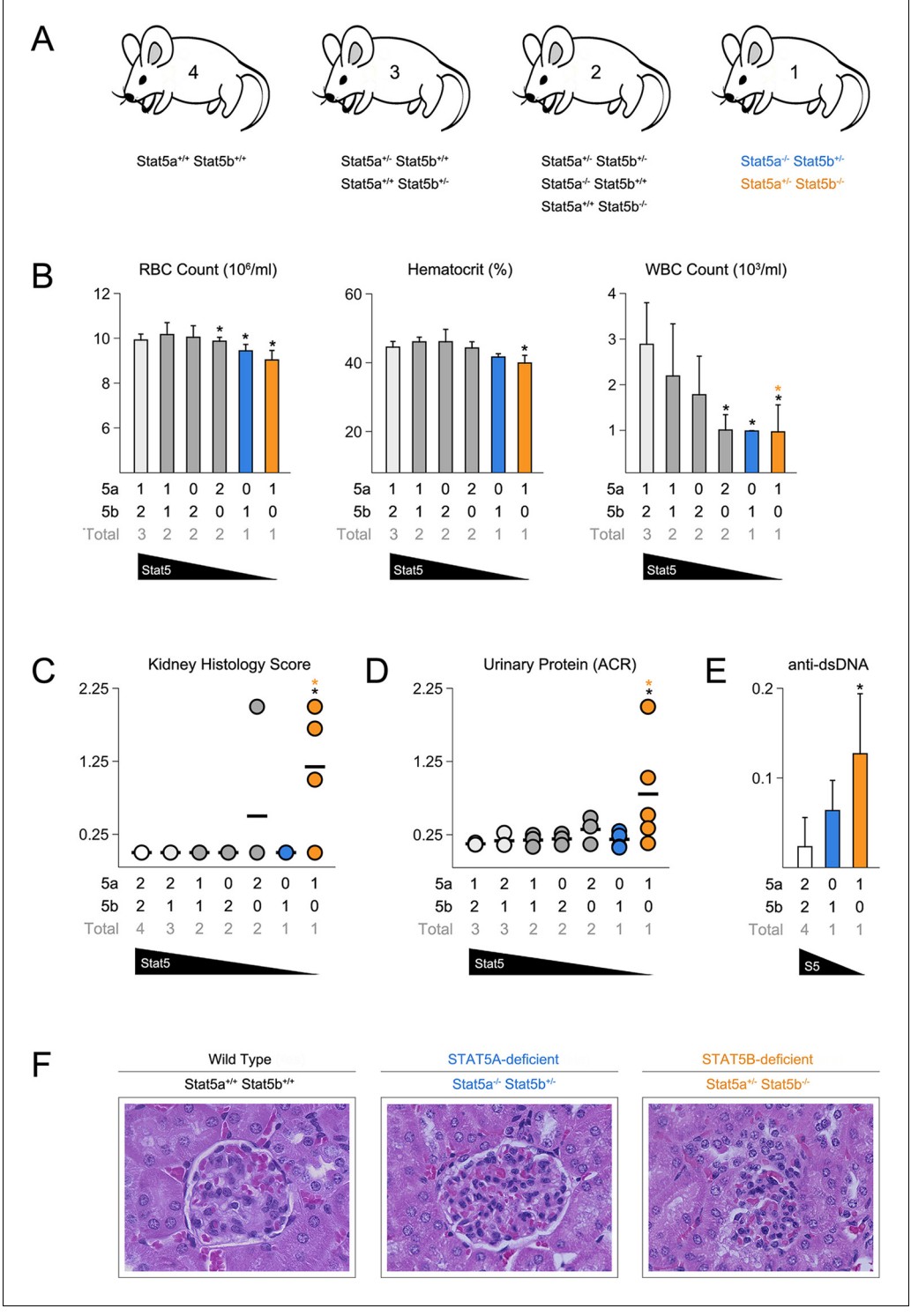

**Figure 1.** Stat5b is required for immunological tolerance. (**A**) Cartoon depicts the mutant mice used in this study. Genotypes are grouped according to total *Stat5* alleles. (**B**) Bar graphs show averaged RBC, hematocrit and WBC counts. (**C**) Scatter plot shows kidney pathology scores. (**D**) Scatter plot shows urinary albumin/creatinine protein ratios. (**E**) Bar graph shows ELISA measurements (O.D.) for anti-double stranded DNA antibodies in serum. (**F**) Micrographs show representative H & E kidney sections (40X magnification). (**B–E**) Number of *Stat5a, Stat5b* and total *Stat5* alleles (i.e. genotype) is explained in the key below each graph. Data are compiled from 3–5 mice per genotype. Error bars indicate standard deviation.

immunological tolerance. Surprisingly, genome-wide DNA binding and transcriptome surveys did not uncover widespread differences in target gene selection but, instead, point towards relative abundance as the key distinguishing factor. Thus, we propose that asymmetric expression (i.e. paralog dose), rather than differential function, determines the dominant STAT5 paralog in lymphoid cells.

## Results

### A dominant role for STAT5B in immunological tolerance

To investigate the relationship between STAT5A and STAT5B, we generated a series of mice with pre-determined combinations of *Stat5* alleles, ranging from two alleles each of A and B (4 total) to one allele of either A or B (*Figure 1A*)(*Yamaji et al., 2013*). We refer to each genotype according to the total number of *Stat5* alleles that are retained. For example, two-allele *Stat5a*-deficient mice lack both *Stat5a* alleles but retain two of *Stat5b* (*Stat5a$^{-/-}$ Stat5b$^{+/+}$*), while one-allele *Stat5a*-deficient mice lack both *Stat5a* alleles and retain just one of *Stat5b* (*Stat5a$^{-/-}$ Stat5b$^{+/-}$*). All 8 genotypes were born at the expected Mendelian ratios and survived beyond 6 months of age, thereby demonstrating that a single allele of either paralog is sufficient to prevent the perinatal lethality seen in STAT5-null mice (**Data not shown**)(*Hoelbl et al., 2006*; *Cui et al., 2004*). Red blood cell counts and hematocrits were comparable across all genotypes, indicating that a single allele is also enough to support erythropoiesis, but white blood cell (WBC) counts were sharply reduced in one-allele *Stat5a*- or

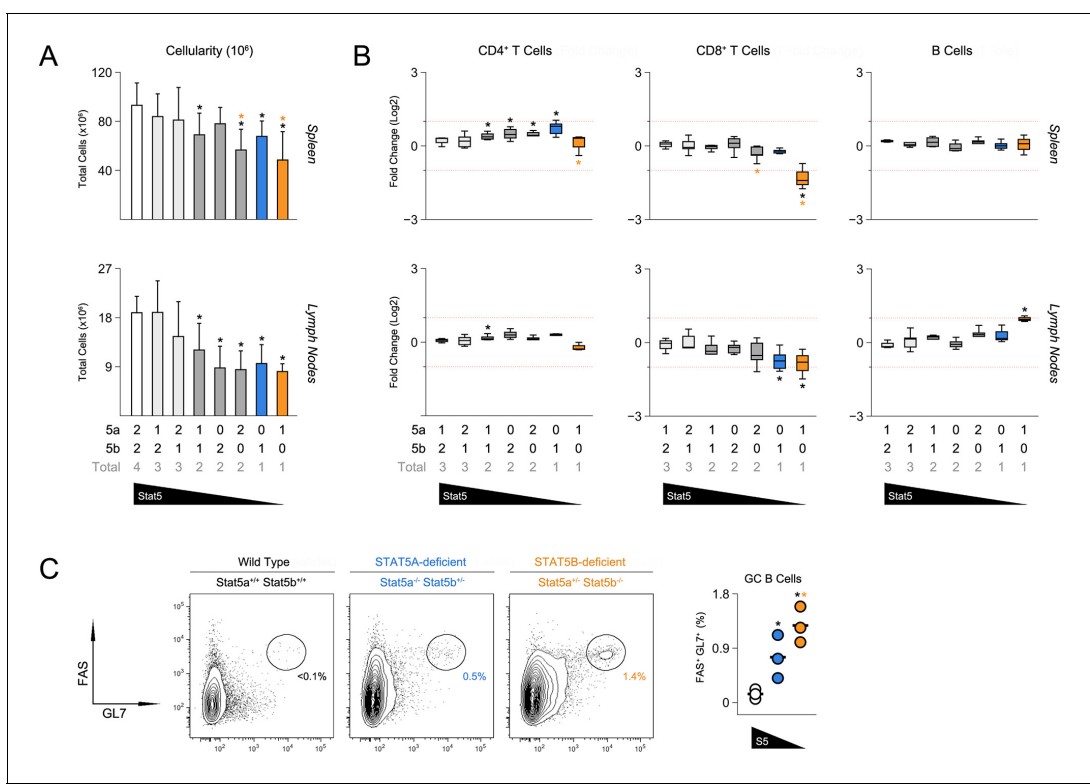

**Figure 2.** Impact of STAT5 paralog deficiency on B and T cells. (**A**) Bar graphs show averaged cell counts for spleens (top row) and lymph nodes (bottom row). Error bars indicate standard deviation. (**B**) Percentages of CD4$^+$ T cells (CD3$^+$ CD4$^+$ CD8α$^-$), CD8$^+$ T cells (CD3$^+$ CD4$^-$ CD8α$^+$) and B cells (CD3$^-$ B220$^+$) were measured in spleens (top row) and LNs (bottom row). Box plots show log2 fold changes relative to wild type controls (WT=0; not shown). Dotted red lines indicate two-fold changes. (**A–B**) Number of *Stat5a*, *Stat5b* and total *Stat5* alleles is explained in the key below each graph. Data are compiled from 5 experiments. (**C**) Contour plots show percentages of GL7$^+$ Fas$^+$ germinal center B cells in lymph nodes. Scatter plot shows percentages of LN resident GC B cells compiled from 3 experiments (3–4 mice per group). Genotypes are ordered as in *Figure 1E* (WT mice = white, one-allele *Stat5a*-deficient mice = blue, one-allele *Stat5b*-deficient mice = orange).

*Stat5b*-deficient mice, as well as two-allele *Stat5b*-deficient mice. By contrast, two-allele *Stat5a*-deficient mice had relatively normal WBC counts (*Figure 1B*).

None of the STAT5 mutants exhibited histological abnormalities in the liver, spleen or intestine, tissues known to be affected in STAT5-null mice (**Data not shown**)(*Snow et al., 2003*; *Yao et al., 2006*). However, *Stat5b*-deficient mice did exhibit kidney pathology with a penetrance of 75% or 25%, depending on whether they harbored one or two *Stat5a* alleles (*Figure 1C*). Afflicted individuals presented a loss of glomerular structure, proteinuria and systemic anti-DNA antibodies (*Figure 1D–F*). Therefore, as in humans, *Stat5b* is required for immunological tolerance in mice but, given the clear difference between having one or two *Stat5a* alleles, redundancy and/or cooperativity is also evident.

## STAT5 paralog dose governs T follicular helper cell responses

To probe for immunological phenotypes, we first assessed the cellularity and composition of primary lymphoid organs. Although not completely lymphopenic like STAT5-null mice (*Yao et al., 2006*), one- and two-allele *Stat5b*-deficient mice did have fewer splenocytes than WT controls (*Figure 2A*). Cell counts were also reduced in one-allele *Stat5a*-deficient mice, suggesting that, while STAT5B may be dominant, STAT5A does have substantial influence. Lymph node cellularity was similarly affected by the loss of either paralog and, in fact, all genotypes with less than three-alleles had reduced cell counts (*Figure 2A*).

Frequencies of CD4$^+$ T cells were comparable across all genotypes, whereas CD8$^+$ T cells were reduced in one-allele *Stat5b*-deficient mice and, to a lesser extent, in one-allele *Stat5a*-deficient mice (*Figure 2B*). By contrast, B cells were increased in one-allele *Stat5b*-deficient mice and, consistent with the appearance of auto-antibodies, GL7$^+$ Fas$^{high}$ IgD$^{low}$ germinal center (GC) B cells were dramatically enriched (*Figure 2B–C* and **data not shown**). One-allele *Stat5a*-deficient mice had a more modest accumulation of GC B cells, again, illustrating both the relevance and redundancy of STAT5A (*Figure 2B–C*).

The ability to promote B cell responses is a defining characteristic of CD4$^+$ 'helper' T cells (*Crotty, 2011*). Therefore, given the appearance of GC B cells, we next investigated the CD4$^+$ T cell compartment. Not surprisingly, there was a marked accumulation of CD44$^{high}$ IL-7R$\alpha^{low}$ effector/memory T cells in *Stat5b*-deficient mice which, as with the incidence of kidney disease, was more pronounced in those bearing one-allele of *Stat5a* than in those bearing two (*Figure 3A-B* & *Figure 3—figure supplement 1A*). We also measured production of IFN-$\gamma$ and IL-17, two effector cytokines that are dysregulated in STAT5-null mice (*Laurence et al., 2007*). IFN-$\gamma^+$ cells were highly enriched in the autoimmune-prone *Stat5b*-deficient mice but not age-matched *Stat5a*-deficient counterparts, suggesting that STAT5B may be particularly important for limiting Th1-type responses. IL-17A$^+$ Th17-type cells were also increased but this trend did not reach statistical significance (*Figure 3C–D*).

CD4$^+$ Tfh cells specialize in promoting B cell responses (*Crotty, 2011*). Mirroring the abundance of GC B cells, there was dramatic accumulation of PD1$^{high}$ CXCR5$^{high}$ ICOS$^{high}$ Tfh cells in one-allele *Stat5b*-deficient mice, and a more modest enrichment in one-allele *Stat5a*-deficient mice (*Figure 3E–F*, *Figure 3—figure supplement 1B* & **data not shown**). At least two interpretations can be made for this disparity; either the two proteins are not functionally equivalent or the two genes have different outputs. Consistent with the latter view, mice lacking one-allele each of *Stat5a* and *Stat5b* (i.e. double-heterozygotes) had more Tfh cells than those lacking two-alleles of *Stat5a*, despite having the same total number of alleles. Moreover, the percentage of Tfh cells was comparable between two-allele *Stat5b*-deficient mice and one-allele *Stat5a*-deficient mice, suggesting that two alleles of *Stat5a* are roughly equal to one allele of *Stat5b* (*Figure 3F* & *Figure 3—figure supplement 1B*).

## STAT5 paralog dose impacts multiple aspects of regulatory T cell function

CD4$^+$ T regulatory (Treg) cells expressing the forkhead transcription factor, FOXP3, are essential for immunological tolerance (*Malek and Castro, 2010*). Given the importance of STAT5 in Treg cells (*Mahmud et al., 2013*), we next inspected this subset. Unlike STAT5-null mice, which exhibit a profound lack of Treg cells (*Yao et al., 2007*; *Burchill et al., 2006*), frequencies of splenic FOXP3$^+$ cells

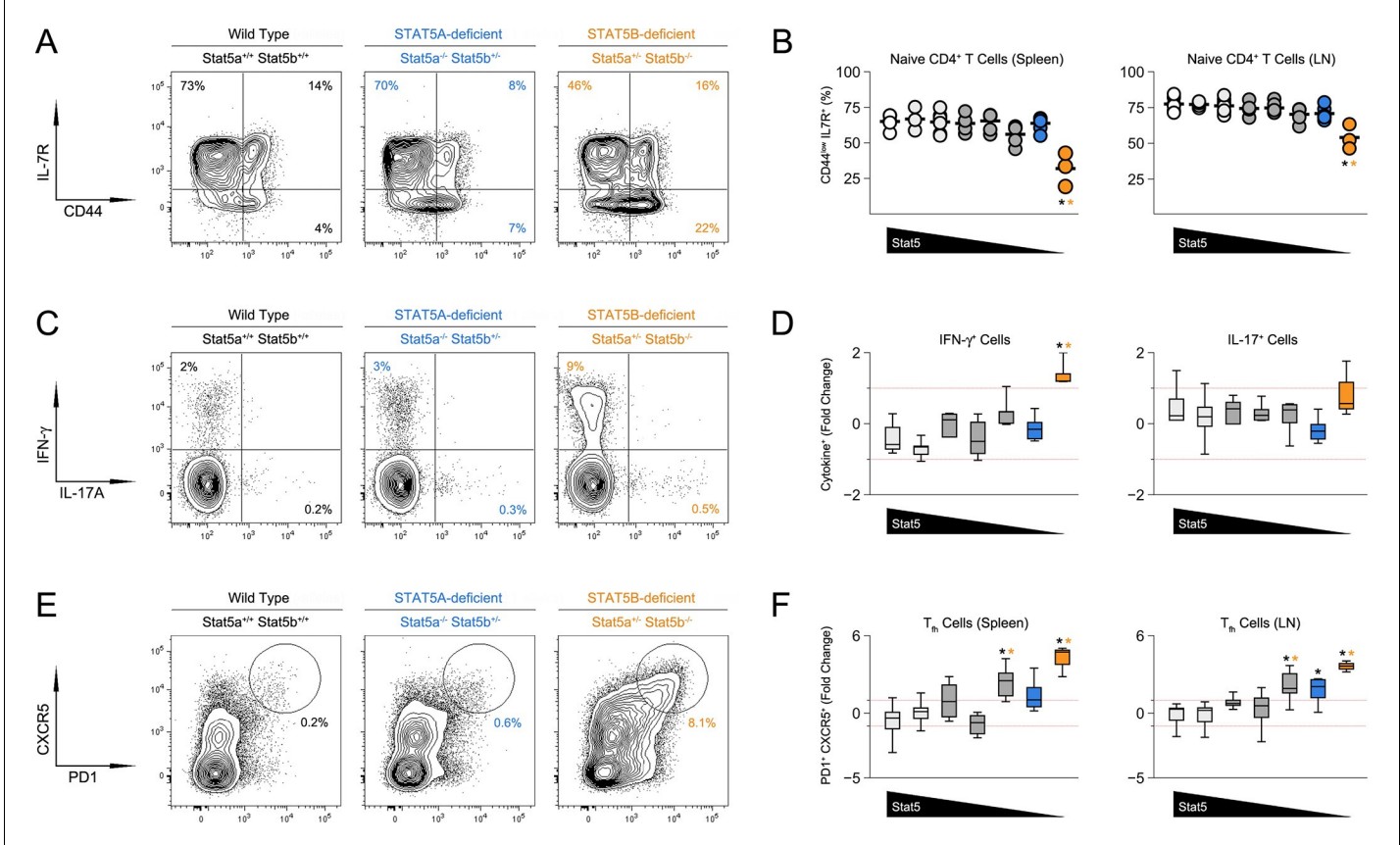

**Figure 3.** Aberrant effector T cell responses in the absence of Stat5b. (A) Contour plots show percentages of CD44<sup>low</sup> IL-7R<sup>+</sup> naive and CD44<sup>high</sup> effector/memory CD4<sup>+</sup> T cells in the spleens of 8 week-old mice. (B) Scatter plots show percentages of naive CD4<sup>+</sup> T cells in spleens (left) and lymph nodes (right). Genotypes are ordered as in *Figure 1C*. (C) Contour plots show percentages of IFN-γ<sup>+</sup> or IL-17<sup>+</sup> CD4<sup>+</sup> T cells in the spleens of 8 week-old mice. (D) Box plots show log2 fold changes for IFN-γ<sup>+</sup> and IL-17<sup>+</sup> cells relative to wild type controls (WT=0; not shown). (E) Contour plots show percentages of PD1<sup>+</sup> CXCR5<sup>high</sup> Tfh cells in the spleens of 8 week-old mice. (F) Box plots show log2 fold changes for Tfh cells in spleens (left) and LNs (right) relative to wild type controls (WT=0; not shown). (D and F) Genotypes are ordered as in *Figure 1D*. Data are compiled from 4 experiments (3–6 mice/group). Dotted red lines indicate two-fold changes.

The following figure supplement is available for figure 3:

**Figure supplement 1.** Impact of Stat5 allele depletion on effector T cell responses.

were relatively normal across our STAT5 mutants. However, due to differences in overall cellularity, absolute counts were significantly lower in one- and two-allele *Stat5b*-deficient mice, as well one-allele *Stat5a*-deficient mice. A similar trend was observed for LN resident Treg cells; frequencies were comparable to WT controls but total numbers were reduced in all genotypes bearing less than 3 total alleles (*Figure 4—figure supplement 1*). To further characterize the Treg compartment, we measured IL-2Rα, a component of the IL-2 receptor that is critical for Treg cell homeostasis and function. It is also a both upstream and downstream of STAT5 signaling and, thus, can be viewed as an indicator of STAT5 activity (*Malek and Castro, 2010*). We found that the percentage of IL-2Rα<sup>+</sup> Treg cells mirrored the total number of *Stat5* alleles; it was slightly reduced in mice with three alleles, lower in those with 2, and lower still in those with 1 (*Figure 4A–B*). We also noted that residual IL-2Rα<sup>+</sup> Treg cells from one-allele mice had reduced suppressive capacity and were unable to maintain expression of IL-2Rα during in vitro culture (*Figure 4C–E*). Each of these phenotypes was more pronounced in the absence of *Stat5b* than *Stat5a*, again, illustrating both the dominance of the former and the relevance (and/or redundancy) of the latter.

Given the appearance of IFN-γ<sup>+</sup> effector T cells in *Stat5b*-deficient mice, we next asked whether *Stat5b*-deficient Treg cells express TBX21, a transcription factor required for Treg cells to limit Th1-

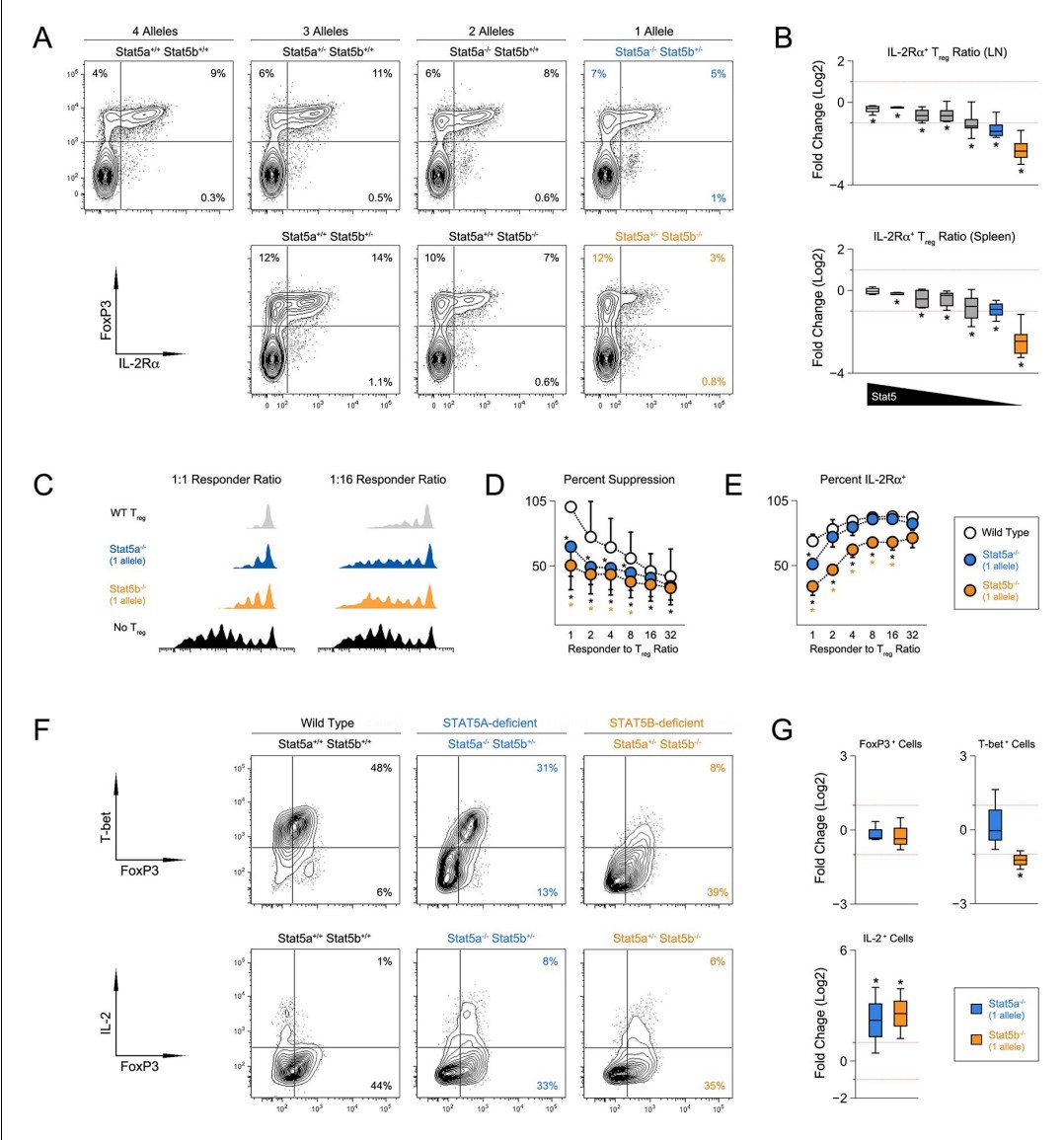

**Figure 4.** T regulatory cell function is impaired in Stat5b-deficient mice. (**A**) Contour plots show percentages of IL-2Rα⁺ cells within the FOXP3⁺ Treg compartment in spleens of 8 week-old mice. (**B**) Box plots show log2 fold changes in the ratio of IL-2Rα⁺/IL-2Rα⁻ Treg (WT=0; not shown). LN (top) and spleen (bottom) data are compiled from 5 experiments (4–6 mice/group) and genotypes ordered as in **Figure 1D**. (**C**) IL-2Rα⁺ Treg cells from WT and *Stat5a-* or *Stat5b*-deficient mice were used for in vitro suppression assays. Histograms show CFSE dilution of responder T cells. (**D**) Line graph shows percent suppression across a range of responder:Treg ratios. Baseline is set according to WT controls at a 1:1 ratio. Data are compiled from 3 experiments. (**E**) Line graph shows the percent Treg cells that remained IL-2Rα⁺ during in vitro suppression. (**F**) IL-2Rα⁺ Treg cells were cultured with IL-2 for 72 hr. Contour plots show the percentage of FOXP3⁺ Treg cells expressing TBX21 (top) or IL-2(bottom). (**G**) Box plots show log2 fold changes for TBX21⁺, FOXP3⁺ and IL-2⁺ cells relative to wild type controls (WT=0; not shown). Data are compiled from 3 experiments. Dotted red lines indicate two-fold changes.

The following figure supplement is available for figure 4:

**Figure supplement 1.** Impact of Stat5 allele depletion on Treg cells.

type responses (*Koch et al., 2009*). Similar to conventional T cells (*Liao et al., 2011*), we found that IL-2 was sufficient to induce TBX21 in WT Treg cells (*Figure 4F*). This effect was slightly reduced in the absence of *Stat5a* but almost completely abolished in the absence of *Stat5b*, consistent with the

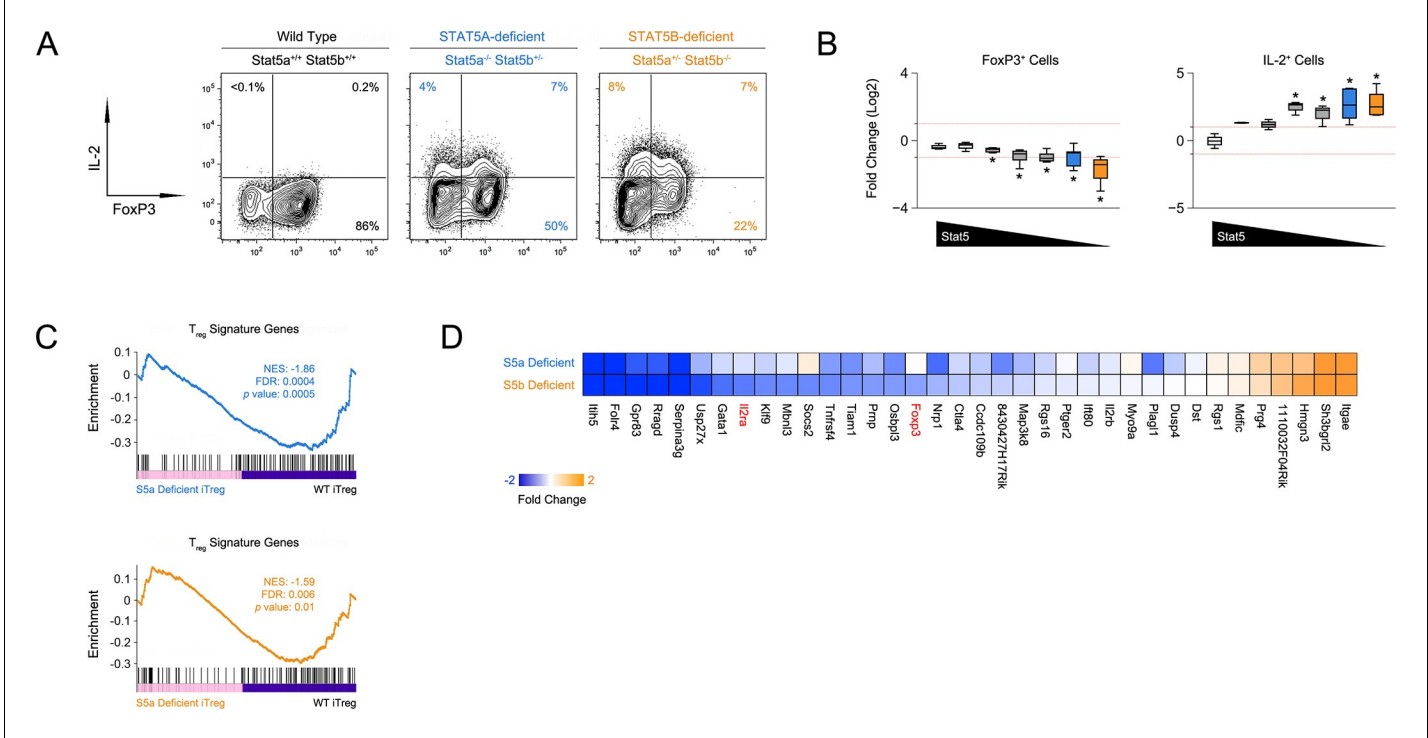

**Figure 5.** Defective iTreg differentiation in the absence of Stat5b. (A) Naive CD4[+] T cells were cultured under iTreg-inducing conditions. Contour plots show percentages of FOXP3[+] and IL-2[+] cells. (B) Box plots show log2 fold changes for FOXP3[+] and IL-2[+] cells relative to wild type controls (WT=0; not shown). Data are compiled from 3 experiments and genotypes ordered as in *Figure 1D*. (C) Naive T cells from one-allele *Stat5a*- or *Stat5b*-deficient mice were cultured as in (A) and processed for RNA-seq. GSEA plots show enrichment of Treg signature genes within the *Stat5a*-deficient (top) or *Stat5b*-deficient (bottom) datasets relative to WT controls. (D) Heat map shows a selection of STAT5-regulated, Treg signature transcripts. Data are presented as log2 fold changes relative to WT controls (not shown). RNA-seq analyses are compiled from 2 biological replicates per genotype.

disparity of other STAT5-dependent parameters (e.g. T cell and Th1 cell frequencies). Both *Stat5a*- and *Stat5b*-deficient Treg cells maintained FOXP3 expression similar to WT controls and, surprisingly, both gained the ability to produce IL-2, a cytokine that is typically restricted in this lineage (*Figure 4F–G*).

STAT5 is required for in vitro differentiation of induced regulatory (iTreg) T cells (*Yao et al., 2007*). To dissect the contributions of STAT5A and STAT5B, we purified naive CD4[+] T cells from our STAT5 mutants, cultured them under iTreg polarizing conditions and compared expression of FOXP3. We found that, although both paralogs appear to play a role, there were far fewer FOXP3[+] cells in one-allele *Stat5b*-deficient cultures than in (*Figure 5A*). We also found that deletion of either paralog endowed FOXP3[+] iTreg cells with the ability to produce IL-2, which suggests that, beyond differentiation, STAT5 may limit the inflammatory potential of this subset (*Figure 5A–B*).

Next, we compared the transcriptomes of *Stat5a*- and *Stat5b*-deficient T cells cultured under iTreg polarizing conditions. Gene set enrichment analysis revealed that the overall Treg gene signature - defined by a combination of FOXP3- and IL-2-dependent transcriptional programs (*Hill et al., 2007*) - was similarly affected in both genotypes, meaning that there were no broad qualitative differences (*Figure 5C*). However, there were obvious quantitative differences; several key genes, including *Foxp3* and *Il2ra*, were more affected by the loss of STAT5B than STAT5A (*Figure 5D*). Thus, while both paralogs can impact Treg cell biology, deletion of *Stat5b* is clearly more disruptive, befitting its dominant station within immunological tolerance.

## Specificity and redundancy of STAT5 paralogs for gene transcription

To define the molecular basis for phenotypic differences between *Stat5a*- and *Stat5b*-deficient T cells, we employed a bioinformatic approach. First, we compared their transcriptomes either directly ex vivo or after acute exposure to STAT5-activating stimuli. The ex vivo set included naive T cells

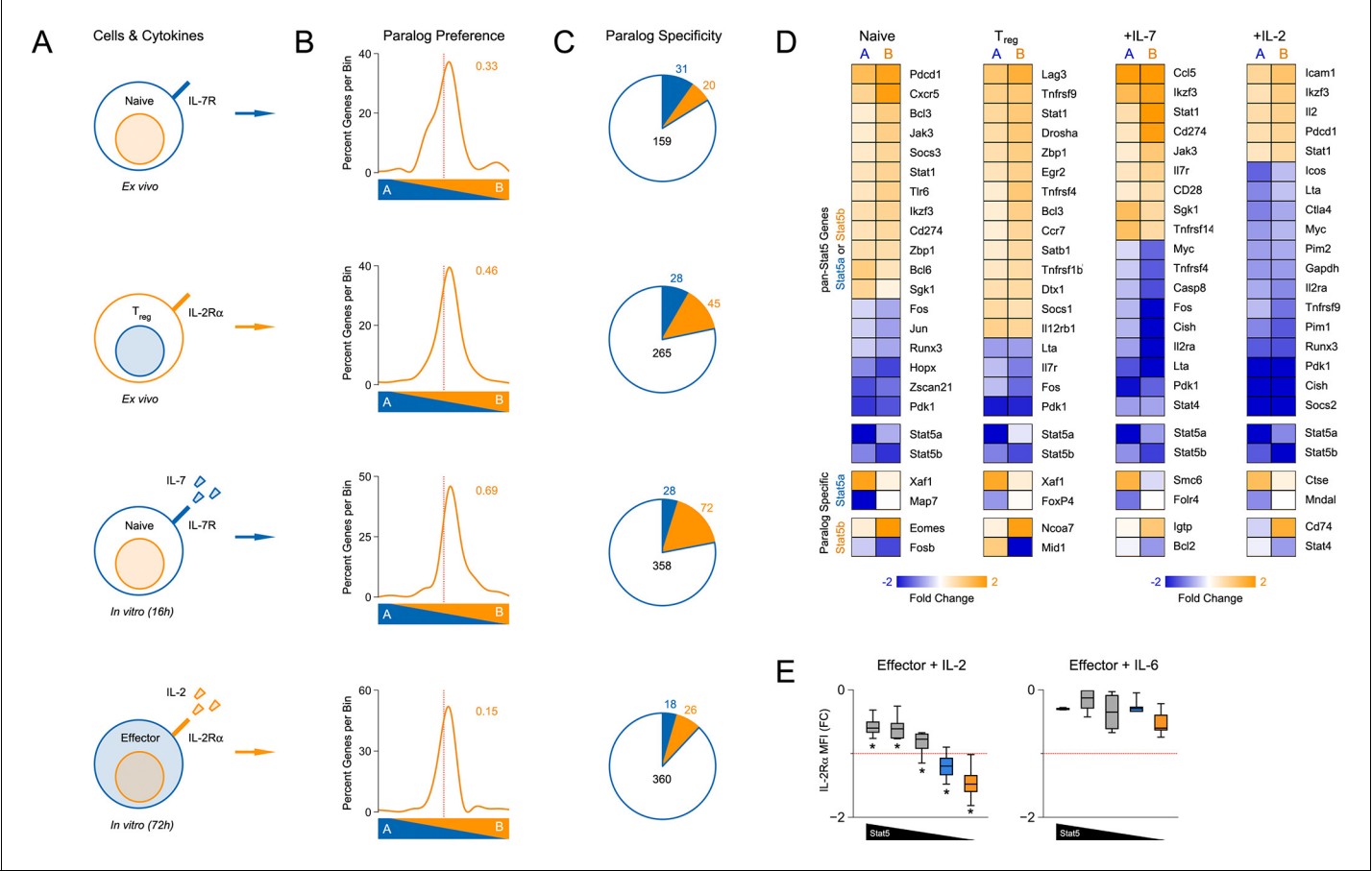

**Figure 6.** *Redundancy and specificity of STAT5 paralogs for gene transcription.* (**A**) Cartoons depict the cell types and experimental conditions used for RNA-seq. (**B**) Histograms show STAT5 paralog preference for all STAT5-regulated transcripts. Those which were more influenced by the loss of *Stat5a* are positioned to the left (X<0) while those that were more influenced by the loss of *Stat5b* are positioned to the right (X>0). Dotted red lines denote equivalence (X=0) and numbers indicate median paralog preference. (**C**) Pie charts depict paralog-specific transcripts. Those impacted only in *Stat5a*-deficient cells are indicated in blue, those impacted only in *Stat5b*-deficient cells are indicated in orange and those impacted in both genotypes are indicated in black. (**D**) Heat maps show a selection of STAT5-regulated transcripts. Data are presented as the log2 fold change relative to WT controls (not shown). (**E**) IL-2Rα protein was measured in T cells treated with IL-2 (left) or IL-6 (right). Box plots show log2 fold changes for mean fluorescence intensity relative to wild type controls (WT=0; not shown). Genotypes are ordered as in *Figure 1D*. Dotted red lines indicate a two-fold change. (**A–D**) RNA-seq analyses are compiled from 2–3 biological replicates per genotype.

The following figure supplements are available for figure 6:

**Figure supplement 1.** Transcriptomic analysis of Stat5a- and Stat5b-deficient T cells.

**Figure supplement 2.** Transcriptomic analysis of Stat5a- and Stat5b-deficient T cells.

andTreg cells, while the in vitro set included naive T cells cultured with IL-7 and effector T cells cultured with IL-2 (*Figure 6A* & *Figure 6—figure supplement 1A*). These pairings were chosen to match the expression patterns of requisite γc co-receptors; IL-7R, which is highly expressed on naive T cells, and IL-2Rα, which is highly expressed on effector T cells (*Rochman et al., 2009*). One-allele mice were used because they exhibited the most dramatic T cell phenotypes.

We began our transcriptomic survey by performing multidimensional scaling of the datasets, thereby gaining a broad overview of the experimental groups. *Stat5a*- and *Stat5b*-deficient cells typically clustered together and equidistant from WT controls, suggesting that the loss of either paralog has comparable genome-wide effects (*Figure 6—figure supplement 1B*). Next, we used statistical variance to identify differentially expressed genes. Surprisingly, we found widespread discord between *Stat5a*- and *Stat5b*-deficient cells; many transcripts appeared dysregulated in the absence

of one paralog or the other (*Figure 6—figure supplement 1C–D*). However, upon close inspection, we concluded that this disparity was largely due to the arbitrary fold-change cutoff that was used. Most genes that were designated as STAT5B-specific were also affected by the loss of STAT5A (and vice versa), albeit to a lesser degree that did not reach our 2-fold threshold (*Figure 6—figure supplement 1E*). To avoid this statistical artifact, we devised a 'paralog preference' scale whereby all STAT5-sensitive genes were compiled and ranked according to how much they were impacted by the loss of *Stat5a* or *Stat5b*. This analysis revealed a binomial distribution for all experimental conditions. The majority of genes were in central bins, affected by both STAT5A and STAT5B, while membership in peripheral bins decreased steadily as paralog preference increased. Importantly, all curves were shifted towards STAT5B, suggesting that STAT5-sensitive genes are generally more impacted by STAT5B than STAT5A (*Figure 6B*). This latter trend was also evident at the protein level; IL-2-driven (but not IL-6-driven) induction of IL-2Rα was clearly more diminished in *Stat5b*-deficient cells than in *Stat5a*-deficient counterparts (*Figure 6E*). Collectively, these data affirm that STAT5B is dominant over STAT5A while, at the same time, demonstrating pervasive redundancy at the level of gene transcription.

Beyond quantitative differences, our transcriptomic survey also revealed qualitative differences between *Stat5a*- and *Stat5b*-deficient cells. Using strict analysis criteria, we discovered that between 12% and 22% of all STAT5-sensitive genes can be classified as paralog-specific, meaning that they are solely dependent on either STAT5A or STAT5B. The absolute number of paralog-specific genes varied across cell states and stimuli, with the largest allotment found in IL-7-treated naive cells, and was typically skewed towards STAT5B (*Figure 6C* & *Figure 6—figure supplement 1E*). Thus, we can create 2 general categories: 'pan-STAT5' genes that are regulated by both STAT5A and STAT5B (e.g. *Pdk1*, *Cish*, *Lta*) and 'paralog-specific' genes that are regulated by either STAT5A (e.g. *Smc6*) or STAT5B (e.g. *Cd74*)(*Figure 6D* & *Figure 6—figure supplement 2*). Given that pan-STAT5 genes are much more numerous, we propose that phenotypic differences between *Stat5a* and *Stat5b* deficient T cells are due largely to paralog preference, owing to the fact that deletion of *Stat5b* has greater quantitative impact, with limited contribution from qualitative, paralog-specific effects.

## Paralog dose dictates genome-wide distribution of STAT5

Functional divergence between STAT5A and STAT5B could be due to differences in target gene selection. Previous studies have addressed this issue by comparing genomic distributions by ChIP-seq (chromatin immunoprecipitation followed by massively parallel sequencing) using separate, paralog-specific antibodies in WT cells (*Liao et al., 2008*; *2011*; *Kanai et al., 2014*). We took an alternative approach involving a single antibody that recognizes both paralogs and T cells from *Stat5a*- or *Stat5b*-deficient mice, as well as 'double heterozygotes' (hereafter referred to as *Stat5a/b^het* mice). In line with previous studies, 1275 unique regions of STAT5 occupancy were called for WT cells(*Figure 7A*). By contrast, *Stat5a/b^het* cells had fewer peaks (658 total; *Figure 7A*) that tended to be less robust (i.e. lower signal intensity) than those found in WT controls (*Figure 7D*), indicating that changes in STAT5 availability can impact genomic distribution even when both paralogs are present. Total peaks were also reduced in *Stat5a*-deficient cells (609 total) and almost expunged in *Stat5b*-deficient cells (97 total), again, illustrating both the relevance of the former and the dominance of the latter (*Figure 7A*). STAT5 peaks were similarly localized across all genotypes - they typically congregated near transcriptional start sites but could also be found at distal regions, sometimes >100 kb from annotated genes - and were highly enriched for STAT-binding motifs (*Figure 7B* & *Figure 7—figure supplement 1*).

Most peaks found in *Stat5b*-deficient cells could be matched to peaks in *Stat5a*-deficient, *Stat5a/b^het* or WT cells (*Figure 7C*). This implies a hierarchy whereby certain sites are preserved even when STAT5B is absent. STAT5B-independent peaks tended to occur near genes that were highly occupied in WT controls and whose expression was highly dysregulated in *Stat5b*-deficient cells (e.g. *Cish*, *Lta*), suggesting that only the most robust (i.e. high-affinity) STAT5-binding sites were preserved (*Figure 8*). Peaks detected within *Stat5a/b^het* and *Stat5a*-deficient cells also tended to be highly occupied in WT controls and dysregulated in STAT5-deficient cells, but the trend was not as dramatic, indicating that, while a full complement of STAT5 alleles may be necessary to achieve optimal responses, STAT5B has the greater influence on genomic distribution (*Figure 8*).

Among the genes that were engaged by STAT5 in WT cells and dysregulated in *Stat5*-deficient cells was *Il2ra*, which, as discussed, is a known STAT5 target gene that is critical for Treg function

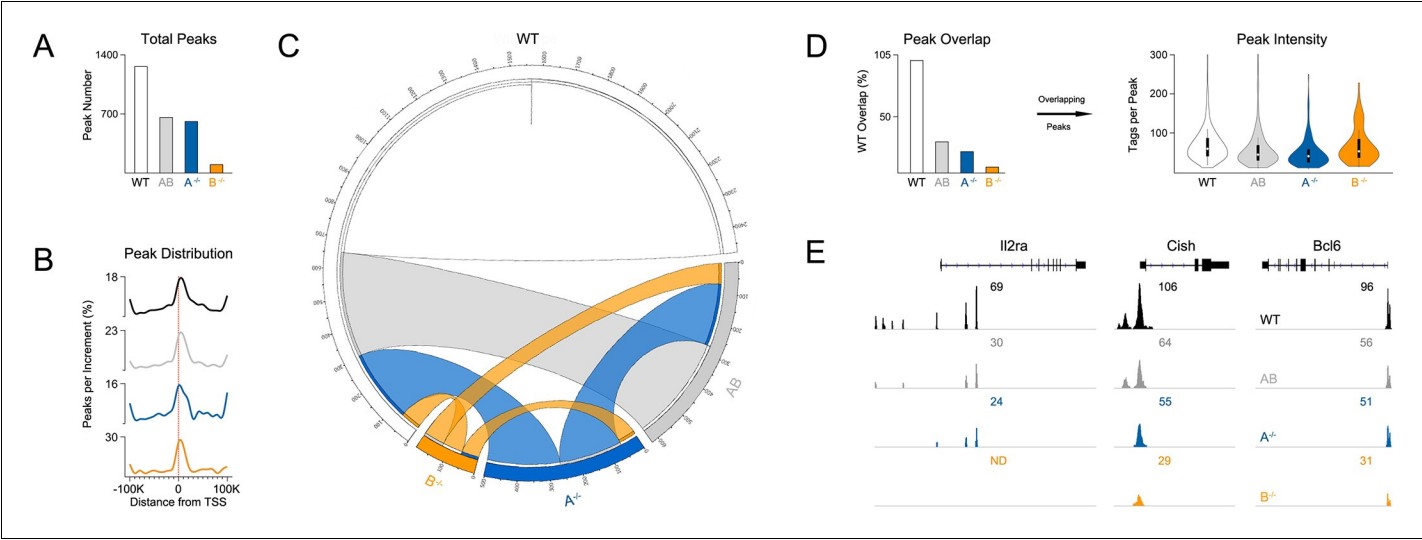

**Figure 7.** Influence of paralog dose on genomic distribution of STAT5. (**A**) CD4[+] T cells from WT, *Stat5a/b*[het] and two-allele *Stat5a-* or *Stat5b*-deficient mice were cultured in the presence of IL-2, then processed for pan-STAT5 ChIP-seq. Bar graph shows the total number of STAT5-bound peaks in each genotype. (**B**) Histogram shows distribution of STAT5-bound peaks relative to transcriptional start sites (TSS). (**C**) Circos plot shows overlap of STAT5-bound beaks across genotypes. Connection width represents the number of overlapping peaks. Only peaks shared with WT cells are shown. Those found only in WT cells are presented as a white semi-circle at the top. (**D**) Bar graph shows the percentage of WT peaks detected in each genotype (WT=100%). Violin plot depicts the total number of sequenced tags (i.e. peak intensity) for peak shared with WT controls. (**E**) Genome browser tracks show STAT5 peaks near selected genes. Numbers indicate the maximum peak height within the interval. (**A–E**) Data are representative of two biological replicates.

The following figure supplements are available for figure 7:

**Figure supplement 1.** Transcription factor motifs associated with STAT5 binding peaks.

**Figure supplement 2.** Correlation between STAT5 binding and transcription of Il2ra, Bcl2 and Bcl6.

and homeostasis (*Figure 7E* & *Figure 7—figure supplement 2*). Another was *Bcl6*, considered the master transcription factor for Tfh differentiation (*Crotty, 2011*)(*Figure 7E* & *Figure 7—figure supplement 2*). In this case, STAT5-binding appears to be a negative regulatory event; multiple studies (including the present work) have shown that STAT5 can suppress *Bcl6* expression in T cells (*Oestreich et al., 2012*; *Liao et al., 2014*). Thus, taken together, our RNA-seq and ChIP-seq data provide a molecular rationale for the Treg and Tfh phenotypes seen in *Stat5*-deficient mice.

## Asymmetric expression of STAT5 paralogs in helper T cells

Based on our RNA-seq and ChIP-seq studies, we reasoned that asymmetric expression, rather than widespread paralog-specific activity, likely explains the phenotypic differences between *Stat5a-* and *Stat5b*-deficient T cells. To explore this possibility, we mined various transcriptome datasets (including our own) and confmed that, indeed, *Stat5b* is more abundant than *Stat5a* at the mRNA level (*Figure 9—figure supplement 1*). Next, we used flow cytometry to measure total STAT5 protein in naive, regulatory (Treg), follicular (Tfh) and effector/memory T cells. Regardless of cellular subset, the results were clear: removing one-allele of *Stat5b* (*Stat5a*[+/+] *Stat5b*[+/-]) had greater impact than removing one-allele of *Stat5a* (*Stat5a*[+/+] *Stat5b*[+/-]) while, at the other end of the spectrum, retaining one-allele of *Stat5b* (*Stat5a*[-/-] *Stat5b*[+/-]) than retaining one-allele of *Stat5a* (*Stat5a*[+/-] *Stat5b*[-/-]) (*Figure 9A*). A similar trend was observed for tyrosine-phosphorylated STAT5 downstream of IL-2 or IL-7 (*Figure 9B*). Thus, we conclude that STATB makes a greater contribution to the total STAT5 protein pool.

To determine how STAT5 availability (i.e. paralog dose) influences gene expression, we transduced *Stat5b*-deficient T cells with a STAT5A-expressing retrovirus, thereby increasing the total amount of STAT5 without re-introducing STAT5B. We first validated the system by measuring *Il2ra*,

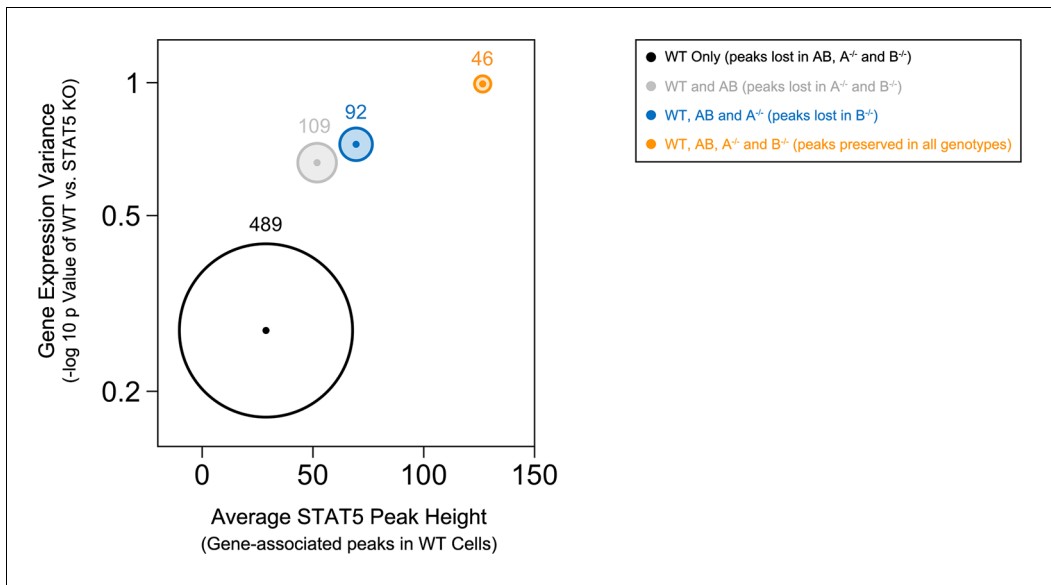

**Figure 8.** Preservation of high affinity targets in the absence of STAT5B. Circle plot relates STAT5 occupancy and STAT5-dependent transcription for genes bound in: (1) only WT cells (white circle), (2) WT and *Stat5a/b^het*, cells (grey circle), (3) WT, *Stat5a/b^het*, and *Stat5a^-/-* cells (blue circle), (4) WT, *Stat5a/b^het*, *Stat5a^-/-* and *Stat5b^-/-* cells (orange circle). STAT5 ChIP-seq peaks were assigned to genes based on proximity to transcriptional start sites (+/-10 kb). X axis denotes the average height of gene-associated peaks in WT cells. Y axis denotes the average mRNA expression variance (-log10 *p* value) for the corresponding peak-associated genes. Variance is derived from the comparison WT and one-allele *Stat5a-* or *Stat5b*-deficient cells cultured in the presence of IL-2 (from *Figure 6*). Size of each circle represents the total number of gene-associated peaks in each group (number is shown).

a well-documented STAT5 target, and found it to be highly induced at both the mRNA and protein levels (*Figure 10A*). Transcriptomic analysis revealed that, overall, ectopic STAT5A mobilized 320 genes, most of which fall within the pan-STAT5 category (e.g. *Cish, Lta*)(*Figure 10A–B*). Applied to our paralog preference scale, these genes did not favor STAT5A, meaning that they were similarly affected in *Stat5a-* and *Stat5b*-deficient cells (*Figure 10A*), and GSEA revealed a high degree of enrichment for both STAT5A- or STAT5B-dependent gene sets (*Figure 10C*). Thus, our data support the idea that differences in STAT5 protein concentrations underlie many (if not most) of the transcriptomic divergence between *Stat5a-* and *Stat5b*-deficient cells.

Having established that ectopic STAT5A can rescue gene expression in *Stat5b*-deficient T cells, we next asked whether it can rescue cellular differentiation. For these studies, naive T cells from wild type, *Stat5a/b^het* or one-allele *Stat5b*-deficient mice were cultured under iTreg polarizing conditions, transduced with either control or STAT5A retrovirus and FOXP3 measured to assess lineage commitment. As expected, FOXP3 was reduced in control-transduced *Stat5a/b^het* cells and almost completely abolished in control-transduced *Stat5b*-deficient cells (*Figure 10D*). However, when ectopic STAT5A was introduced, the percentage of FOXP3+ cells became comparable across all genotypes and, whether endogenous (top row) or ectopic (bottom row), there was a clear linear correlation between STAT5 and FOXP3 protein levels (*Figure 10D*). IL-2Rα was also diminished in both *Stat5a/b^het* and *Stat5b*-deficients cells, and was restored by ectopic STAT5A (*Figure 10E*). These data argue that a threshold concentration of STAT5 must be reached to institute the Treg program and, given the conspicuous effect of ectopic STAT5A on WT cells (*Figure 10D* & *Figure 10— figure supplement 1A*), they imply that STAT5 is a limiting resource for this process.

Although they share a common instructive cytokine (TGF-β), Th17 cells and Treg cells have opposing pro- and anti-inflammatory functions. STAT5 is key to this divergence - it promotes Treg responses at the expense of Th17 responses – so we next investigated the effect of paralog dose on Th17 differentiation. We found that the percentage of IL-17+ cells was 4-fold higher in *Stat5a/b^het* Th17 cultures and >25-fold higher in *Stat5b*-deficient Th17 cultures than in WT controls, consistent with a high paralog dose threshold, and most importantly, that ectopic STAT5 not only extinguished

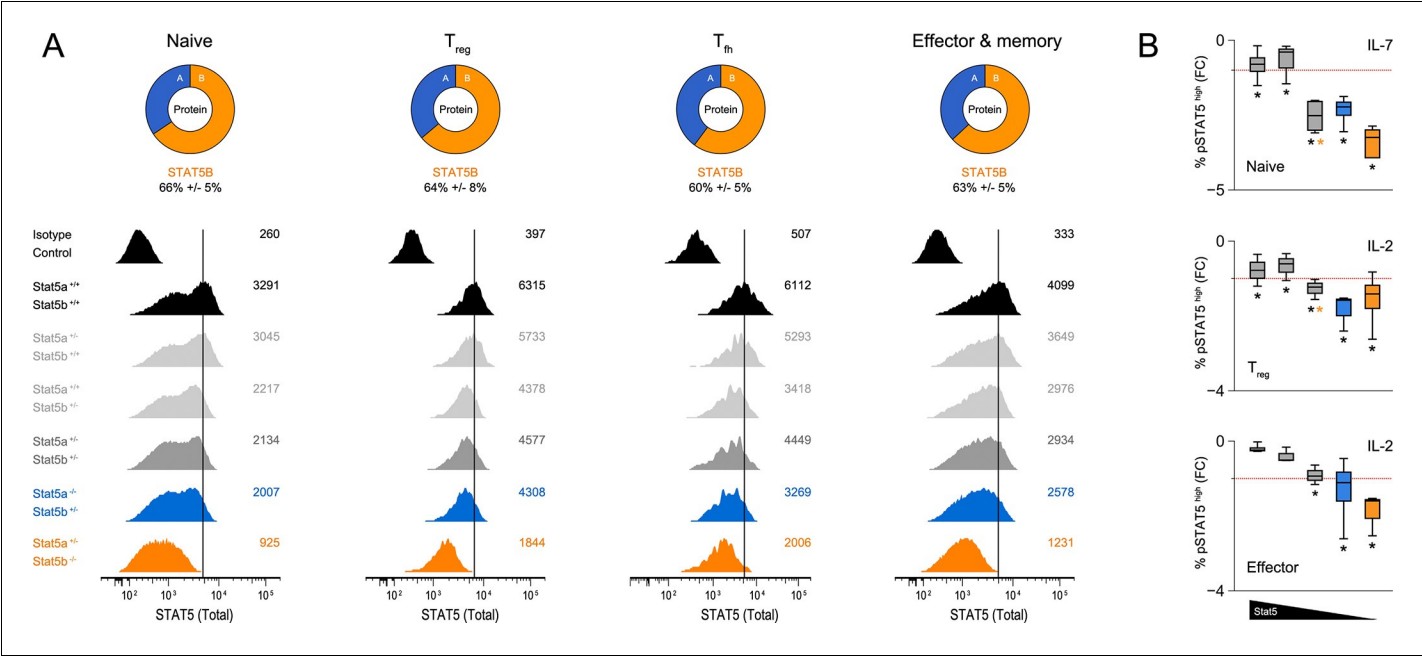

**Figure 9.** Relative abundance of STAT5A and STAT5B in helper T cells. (**A**) Total STAT5 protein was measured in naive, Treg, Tfh and effector/memory T cells. Donut charts indicate the percentage of total STAT5 protein accounted for by each paralog. Histograms show representative flow cytometry data from one of three experiments. (**B**) Naive, Treg and effector/memory T cells were treated with IL-2 or IL-7 and phospho-STAT5 measured by flow cytometry. Box plots show log2 fold changes for the percentage of p-STAT5$^{high}$ cells relative to wild type controls (WT=0; not shown). Genotypes are ordered as in *Figure 1D*. Dotted red lines indicate a two-fold change (3–4 replicates/group).

The following figure supplements are available for figure 9:

**Figure supplement 1.** Relative abundance of STAT5A and STAT5B in helper T cells.

**Figure supplement 2.** Stat5a and Stat5b are transcribed from opposite DNA strands.

**Figure supplement 3.** Putative lymphocyte-restricted enhancers within the Stat5a/b locus.

IL-17 but also induced FOXP3 in all genotypes, thereby demonstrating that changes in STAT5 concentration can tip the balance between effector and regulatory T cells programs (*Figure 10—figure supplement 1B*).

## Discussion

Although the importance of STAT5 is widely recognized, there is no consensus on whether its closely related paralogs, STAT5A and STAT5B, are redundant or functionally distinct. Assuming the latter, it is also unclear how specificity can be achieved given their extensive structural homology. Both positions are grounded in sound experimental evidence but, until the present studies, there has been no comprehensive inquiry on their relationship in immune cells. We addressed this longstanding question in primary CD4$^+$ helper T cells, the principal orchestrators of adaptive immunity. Using a combination of genetic and genomic approaches, we demonstrate that STAT5B is dominant over STAT5A and, thus, plays a non-redundant role in controlling effector and regulatory T cell responses. This conclusion is based on phenotypic differences between *Stat5a-* and *Stat5b*-deficient mice, as well as bioinformatic analyses showing that STAT5B has greater impact on both selection and transcription of STAT5 target genes. The disparity does not appear to be due to differences in genome wide distribution or transcriptional capacity but, instead, relates to differences in relative abundance. Consistent with the latter point, our loss- and gain-of-function studies demonstrate that a threshold concentration of STAT5 must be reached to execute STAT5-dependent gene expression and

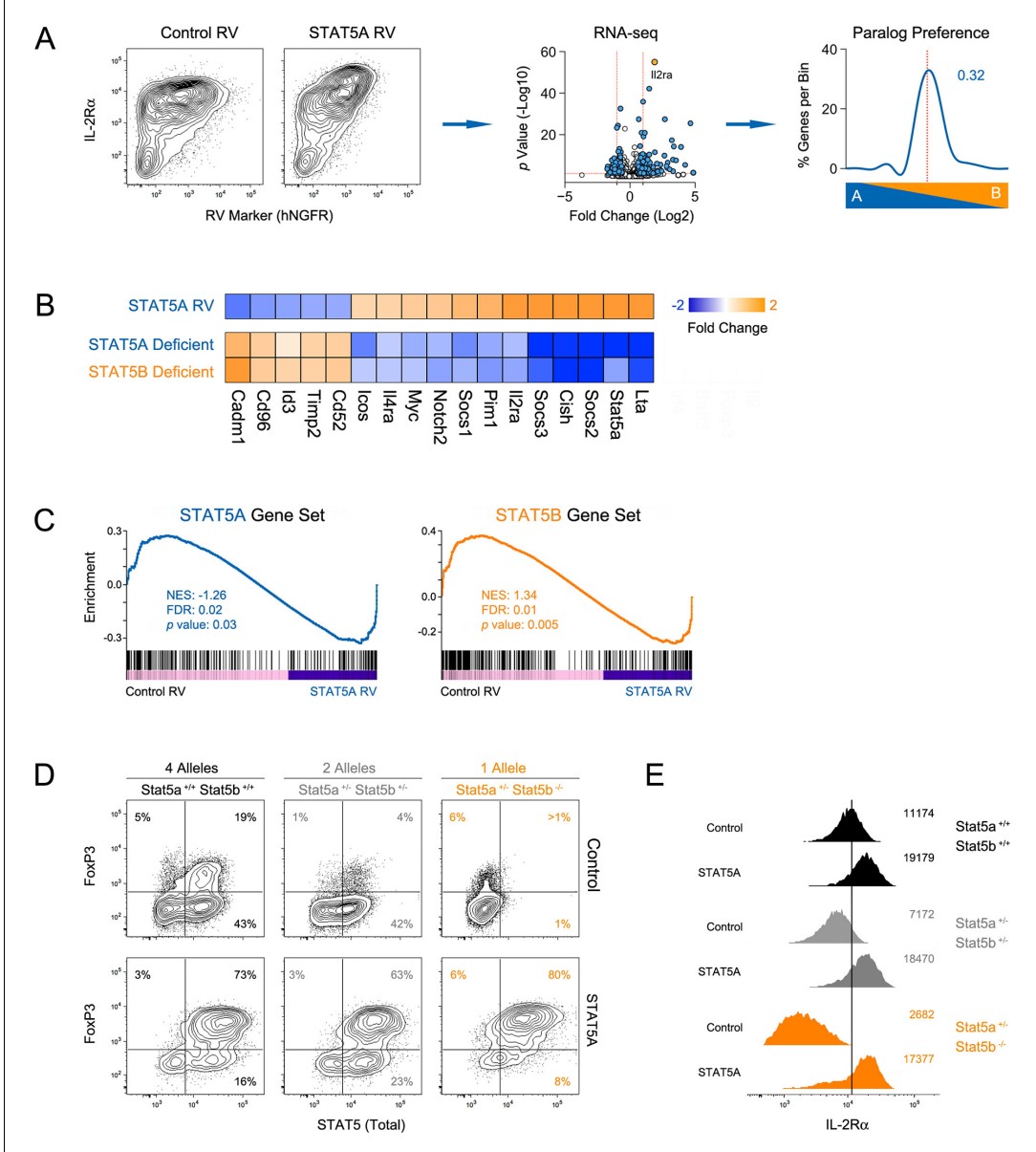

**Figure 10.** Paralog dose governs STAT5-driven gene transcription. (**A**) CD4[+] T cells from one-allele *Stat5b*-deficient mice were transduced with STAT5A retrovirus, then processed for RNA-seq. Contour plots (left) show correlation between the transduction marker (hNGFR) and IL-2Rα protein. Volcano plot (middle) shows log2 fold changes and variances for all transcripts relative to control retrovirus. Those exhibiting >1.5 fold change and p<0.05 are depicted in blue. Dotted red lines indicate 2 fold change and 0.05 *p* value. Histogram (right) shows STAT5 paralog preference for transcripts mobilized by ectopic STAT5A. Dotted red line denotes equivalence and number indicates median paralog preference. (**B**) Heat map shows selected transcripts in STAT5A-transduced helper T cells (top row) or IL-2 treated *Stat5a*- or *Stat5b*-deficient cells (bottom rows; from *Figure 6*). Data are presented as the log2 fold change relative to controls (not shown). (**C**) GSEA plots show enrichment of STAT5A-dependent (left) or STAT5B-dependent (right) genes within the STAT5A-RV dataset. (**A–C**). RNA-seq analysis is compiled from 2 biological replicates. (**D**) CD4[+] T cells from WT, *Stat5a/b^het* and one-allele *Stat5b*-deficient mice were transduced with control (top row) or STAT5A (bottom row) retrovirus under iTreg polarizing conditions. Contour plots show total STAT5 and FOXP3 protein levels in transduced cells. (**E**) Histograms denote IL-2Rα protein levels on transduced cells. (**D–E**) Shown is one of two independent experiments.

The following figure supplement is available for figure 10:

**Figure supplement 1.** STAT5 paralog dose tips the balance between effector and regulatory T cell programs.

differentiation programs. Based on these findings, we submit that STAT5A and STAT5B are largely redundant at the molecular level, but not at the cellular or organismal levels, where STAT5B is dominant.

It has been proposed that the target repertoires of STAT5A and STAT5B vary due to subtle differences in their DNA-binding domains (*Boucheron et al., 1998*). However, this notion has been disputed because the nature and location of the amino acid substitutions may not alter protein structure enough to impact specificity. In addition, multiple studies have shown that the consensus DNA-binding motifs for STAT5A and STAT5B are identical, although it should be noted that these measured optimal binding to synthetic oligonucleotides in cell free systems, leaving open the possibility that divergent binding properties become apparent only at lower affinity sites or in the context of native chromatin (*Soldaini et al., 1999*; *Ehret et al., 2001*). Indeed, differential binding of STAT5A or STAT5B has been detected at several loci in primary immune cells but it remains unclear whether this reflects *bonafide* differences in specificity or other factors that may influence target gene selection (*Liao et al., 2008*; *2011*; *Yamaji et al., 2013*; *Kanai et al., 2014*). For instance, it is known that STAT5A and STAT5B can exhibit distinct phosphorylation patterns, so preferential binding may reflect cell type- or stimulus-specific differences in activation rather than distinct targeting capabilities (*Caldenhoven et al., 1998*; *Hennighausen and Robinson, 2008*; *Rosen et al., 1996*; *Meinke et al., 1996*). Our work supports this latter view by establishing that, even before activation, relative abundance of STAT5A versus STAT5B determines which paralog will dominate a given transcriptional response.

Recent work has shown that small oscillations in transcription factor availability can have genome-wide consequences (*Brewster et al., 2014*). The idea that STAT5 concentration can impact cellular function also has precedent. Of particular interest are studies reporting severe immunological phenotypes in transgenic mice which over-express STAT5A or STAT5B (*Kelly et al., 2003a*; *Kelly et al., 2003b*), and studies showing that *Stat5a/b* haplo-insufficiency ameliorates contact hypersensitivity (*Nivarthi et al., 2014*). In addition, we have previously demonstrated that ectopic STAT5A can expand the target repertoire of STAT5 in mouse embryonic fibroblasts (*Zhu et al., 2012*), and have explored the concept of STAT5 gene dosage in the context of mammary development, finding that a high STAT5 threshold must be reached for mammary epithelial cell differentiation (*Yamaji et al., 2013*). Given that STAT5A is the dominant paralog in mammary epithelium, we can infer that asymmetric expression of STAT5 paralogs is not just a feature of immune cells, and that it must be controlled in a tissue-specific manner (*Metser et al., 2016*). Several mechanisms may explain this phenomenon. First, it is known that *Stat5a* and *Stat5b* are transcribed from opposite DNA strands and, thus, may be subject to strand-specific modes of regulation (*Figure 9—figure supplement 2*). Second, differential transcription could be achieved through paralog-specific enhancer elements whose accessibility is tissue- and/or cell type-restricted. We have recently characterized an intergenic enhancer that drives expression of *Stat5a* in mammary epithelium and have identified multiple DNase hypersensitivity sites within the *Stat5b* locus that are present in T cells but not in non-lymphoid tissues, perhaps indicating an analogous mechanism for immune cells (*Metser et al., 2016*) (*Figure 9—figure supplement 3*). Third, paralog-specific epigenetic modifications, such as histone or DNA methylation, may impose distinct transcriptional outputs, as shown for tumor cells (*Zhang et al., 2007*). Fourth, the 3' UTRs of *Stat5a* and *Stat5b* are highly divergent so it is possible that their mRNAs are subject to post-transcriptional regulation via distinct sets of microRNAs and/or RNA-binding proteins (*Liu et al., 1995*).

Beyond asserting the dominance of STAT5B, our work also affirms the importance of STAT5A. Several observations support this latter point: 1) deletion of one *Stat5a* allele exaggerates the gross and cellular phenotypes of *Stat5b*-deficient mice, 2) transcription of STAT5 target genes is typically influenced by both STAT5A or STAT5B, and 3) ectopic STAT5A can rescue gene expression in *Stat5b*-deficient cells. Furthermore, just one-allele of either *Stat5a* or *Stat5b* is sufficient to prevent the perinatal lethality and anemia seen in STAT5-null mice, suggesting that molecular redundancy protects the most critical 'life-and-death' functions. We also identified a small subset of genes that appear to be regulated by either STAT5A or STAT5B, some of which have known immunological functions. Given that STAT5B is more abundant, it can be argued that all STAT5B-dependency may be due to a high paralog dose threshold, but this cannot explain the appearance of STAT5A-dependent genes. Thus, we propose that phenotypic differences between *Stat5a*- and *Stat5b*-deficient T cells result from widespread 'paralog preference' and circumscribed 'paralog specificity'.

Our ChIP-seq studies indicate that the overall availability of STAT5, whether STAT5A or STAT5B, has profound influence on target gene selection. Previous studies have compared genomic distribution of STAT5A and STAT5B in primary CD4[+] T cells and found that they mostly overlap, thereby supporting the idea of redundancy (*Liao et al., 2008*; *2011*; *Kanai et al., 2014*). However, they also identified a subset of sites that are occupied by one paralog or the other and, thus, have been taken as evidence for paralog specificity. All such comparisons (including ours) should be interpreted with care. Shared sites can be appointed with confidence but, due to technical confounders (e.g. differences in antibody affinity), incongruent sites cannot be definitively classified as STAT5A- or STAT5B-specific. The mechanisms underlying differential binding must also be considered. It is possible that *bona fide* paralog-specific binding sites do exist, but these are probably only a minor fraction. In most cases, differential binding likely reflects competition; the more abundant paralog is more likely to be detected. Given this nuance, claims that certain genes are uniquely regulated by STAT5A or STAT5B should be tempered. For instance, it has been suggested that *Bcl2l1* is regulated only by STAT5A and that *Bcl2*, *Il2ra* and *Foxp3* are regulated only by STAT5B (*Kanai et al., 2014*; *Jenks et al., 2013*). Our data indicate that these are more accurately described as 'pan-STAT5' genes that are more impacted by deletion of one paralog or the other.

STAT5 is essential for immunological tolerance (*Mahmud et al., 2013*). This principle is well illustrated in humans with congenital *STAT5B* defects, who typically manifest a range of autoimmune symptoms (*Kanai et al., 2012*), and is further supported by the present work, which demonstrates that *Stat5b* deficiency leads to spontaneous kidney disease in mice. The link between STAT5 and autoimmunity is often attributed to its role downstream of IL-2/IL-2Rα in Treg cells (*Malek and Castro, 2010*; *Mahmud et al., 2013*). Our work clearly endorses this viewpoint and brings to mind the autoimmune phenotype of Treg-deficient *Scurfy* mice which, like *Stat5b*-deficient mice, exhibit both autoantibodies and kidney disease {Aschermann:2013gn}. We demonstrate that, similar to *STAT5B*-deficient humans (*Cohen et al., 2006*), Treg cells are functionally compromised in *Stat5b*-deficient mice, but, surprisingly, the baseline frequency of FOXP3[+] cells was not reduced, likely reflecting immunological and/or environmental differences between the two species. We also present new ideas about why STAT5-deficient Treg cells are impaired. First, they acquire the ability to produce IL-2, a cytokine that is typically restricted in Treg cells (*Malek and Castro, 2010*). This finding is consistent with previous studies demonstrating that STAT5 can suppress IL-2 production by conventional T cells and, given that Treg cells are thought to operate, in part, by consuming IL-2, it provides one explanation for their ineffectiveness (*Villarino et al., 2007*; *Pandiyan et al., 2007*). Second, they fail to express TBX21, a transcription factor that is required to limit Th1-type T cell responses (*Koch et al., 2009*). Previous studies have shown that STAT1-activating cytokines (e.g. interferons, IL-27) can induce TBX21 in Treg cells but we are the first to show that IL-2, a STAT5-activating cytokine, can do it (*Hall et al., 2012*; *Koch et al., 2012*).

Aside from its role in Treg cells, STAT5 promotes immunological tolerance via effector cell-intrinsic mechanisms. Given the dramatic accumulation of Tfh cells in our STAT5 mutants, the capacity of Tfh cells to promote autoimmunity, and recent work showing that STAT5 can suppress Tfh differentiation (*Ballesteros-Tato et al., 2012*; *Johnston et al., 2012*), we conclude that exaggerated Tfh responses factor heavily in the autoantibody responses and attendant kidney pathology seen in *Stat5b*-deficient mice. Our data also suggest an intimate relationship between STAT5 and BCL6, the 'master' transcription factor for Tfh cells. We report that STAT5 directly engages the *Bcl6* locus, where it likely acts as a transcriptional repressor, and that STAT5 binding sites are often enriched for BCL6 motifs, consistent with published accounts of co-localization between these two transcription factors (Y. *Zhang et al., 2012*; *Liao et al., 2014*). These findings strongly implicate Tfh cells in the pathogenesis of *Stat5b*-deficient mice but, since these are germline 'knockouts', we must consider the (likely) possibility that intrinsic defects in other cell types contribute to the autoimmune phenotype. For instance, multiple dendritic cell subsets are known to be dysregulated in *Stat5*- or *Jak3*-deficient mice (*Esashi et al., 2008*; *Yamaoka, 2005*), and its influence on non-immune cells, particularly downstream of hormone receptors, cannot be ignored (*Kuhrt and Wojchowski, 2015*; *Hennighausen and Robinson, 2008*).

Because of its prominent role within the immune system, STAT5 has long been viewed as an attractive target for therapeutic intervention. Clinical use of STAT5-activating cytokines and growth factors (e.g. IL-2, erythropoietin) is now commonplace and the recent approval of Jak3 inhibitors for the treatment of autoimmune disease and malignancy points to sustained interest in this pathway

(*Villarino et al., 2015*). Consequently, a detailed understanding of how STAT5 signaling works is imperative not only to inform new drugs, but also to improve existing regimens. The present study yields multiple clinically relevant insights and, in particular, raises two key issues that should be considered. First, partial inhibition of STAT5 expression and or activity may be sufficient to have desired effects on immune cell function. Second, targeting of STAT5A may be safer (though perhaps less robust) than targeting of STAT5B. Therefore, taking a broad view, our findings provide a molecular rationale for exploiting STAT5 paralog redundancy in clinical settings.

## Materials & methods

### Experimental Animals

STAT5 mutants were generated as described (*Yamaji et al., 2013*). Briefly, mice lacking the entire *Stat5* locus ($Stat5a/b^{+/-}$) were crossed with mice lacking one-allele of *Stat5a* ($Stat5a^{+/-}$ $Stat5b^{+/+}$) or *Stat5b* ($Stat5a^{+/+}$ $Stat5b^{+/-}$) to produce 8 combinations of *Stat5* alleles (*Figure 1A*). We refer to each genotype according to the total number of *Stat5* alleles that are retained. For example, two-allele *Stat5a*-deficient mice lack both *Stat5a* alleles but retain two *Stat5b* alleles ($Stat5a^{-/-}$ $Stat5b^{+/+}$), while one-allele *Stat5a*-deficient mice lack both *Stat5a* alleles but retain one *Stat5b* allele ($Stat5a^{-/-}$ $Stat5b^{+/-}$). CD45.1$^+$ C57BL/6 mice were purchased from Jackson Labs (Bar Harbor, ME). Animals were handled in accordance with NIH guidelines and all experiments approved by the NIAMS Animal Care and Use Committee.

### Blood, urine and lymphoid tissue analysis

Complete blood counts were taken from 8 to 12-week old mice (NIH Clinical Center, Division of Veterinary Services, Bethesda, MD). Anti-double stranded DNA antibodies were measured in serum collected from 4 to 6 month old mice (Calbiotech, Spring Valley, CA). Albumin/creatinine ratio was measured in urine collected from 4 to 6 month old mice (Exocell, Philadelphia, PA). Spleen and lymph node (cervical, axillary, brachial and inguinal) cellularity was measured in 8–12 week old mice using a Nexcelom X1 Cellometer (Lawrence, MA).

### Histology

Kidneys were dissected from 4 to 6-month old mice, fixed, embedded in paraffin, sectioned and stained with haematoxylin and eosin (American Histolabs, Gaithersburg, MD). Blinded scoring was performed by a veterinary pathologist (Diagnostic & Research Services Branch, National Institutes of Health, Bethesda, MD). Specimens from at least 3 mice per genotype were inspected. Micrograph images were collected using a BioRevo BZ-9000 digital microscope (Keyence, Itasca, IL).

### Flow cytometry

For surface proteins, cells were stained directly ex vivo with fluorochrome labelled anti-mouse CD3ε, CD4, CD8α, CD25 (IL-2Rα), CD44, CD45R (B220), CD95 (FAS), CD127 (IL-7R), CD185 (CXCR5), CD279 (PD1), GL-7, and IgD. For intracellular proteins, cells were fixed and permeabilized using transcription factor staining buffer set (eBioscience, San Diego, CA), then stained with fluorochrome labelled anti-mouse FOXP3 and/or TBX21. For cytokine production, cells were stimulated with Phorbol 12-myristate 13-acetate and ionomycin for 4 hr (50 ng/ml and 500 ng/ml, respectively; Sigma-Aldrich, St. Louis, MO), treated with Brefeldin A for 2 hr (10 µg/ml; Sigma-Aldrich), fixed (2% formaldehyde; Sigma-Aldrich), permeabilized (0.25% Saponin; Sigma/Aldrich), and stained with fluorochrome-labelled anti-mouse IFN-γ, IL-2 and/or IL-17A. For IL-2Rα induction, naive CD4$^+$ CD44$^{low}$ CD25$^-$ cells were purified from pooled lymph nodes and spleens using a FACS Aria Cell Sorter (>98% purity; BD Biosciences, San Diego, CA). These were stimulated with plate-bound anti-CD3 (10 µg/ml; Clone 17A2) and soluble anti-CD28 (1 µg/ml; Clone 37.51) in the presence of soluble anti-mouse IL-2, IL-4 and IFN-γ (10 µg/ml each; Clones S4B6, BVD6-24G2 and XMG1.2; BioXcell, West Lebanon, NH) for 18 hr, then treated with human IL-2 (100 units/ml; NIH/NCI BRB Preclinical Repository) or mouse IL-6 (20 ng/ml; eBioscience) for 18 hr and stained with fluorochrome labelled anti-mouse CD25. For tyrosine-phosphorylated STAT5, splenocytes were treated directly ex vivo with human IL-2 (100 units/ml) or mouse IL-7 (20 ng/ml; eBioscience) for 1 hr, or stimulated with anti-CD3 and anti-CD28 in the presence of anti-mouse IL-2 for 18 hr, then pulsed with human IL-2 for

1 hr (100 units/ml). These were then fixed with 2% formaldehyde, permeabilized with 100% methanol and stained with Alexa Fluor 647-labelled anti-human/mouse pY694 STAT5 (Clone 47; BD Biosciences) in conjunction with fluorochrome labelled anti-mouse CD3ε, CD4, CD25, CD44, CD127 and/or FOXP3. Total STAT5 protein was measured in splenocytes directly ex vivo or following retroviral transduction of purified CD4$^+$ T cells (described below). In both cases, cells were fixed with 2% formaldehyde, permeabilized with 100% methanol, then stained with a rabbit polyclonal IgG that recognizes both STAT5A and STAT5B (sc-835; Santa Cruz Biotechnology, Santa Cruz, CA) in conjunction with fluorochrome labelled anti-mouse CD3ε, CD4, CD25, CD44, CD127, (IL-7R), CD185 (CXCR5), CD279 (PD1), IL-17A and/or FOXP3. Phycoerythrin-labelled goat anti-rabbit IgG was used for detection (ac-3739; Santa Cruz Biotechnology). Normal rabbit IgG was used as a negative control (ac-2027; Santa Cruz Biotechnology).

All fluorochrome-labelled antibodies were purchased from eBioscience, BD Biosciences or Biolegend (San Diego, CA), unless noted otherwise. Data were collected on a FACSverse cytometer (BD Biosciences) and analyzed using FlowJo software (FlowJo LLC, Ashland, OR). Compiled cytometry data are presented as scatter plots where each element represents a single replicate (horizontal line indicates the mean), or box plots where the the fold change for each replicate was calculated relative to WT controls and log 2 transformed (horizontal line indicates the mean and whiskers indicate minimum and maximum values). Cells were maintained in supplemented tissue culture medium (RPMI-1640 with 10% fetal calf serum, 1% sodium pyruvate, 1% nonessential amino acids, 0.1% β-Mercaptoethanol, 100 U/ml penicillin, 100 mg/ml streptomycin; Life Technologies, Grand Island, NY) and cultured at a density of 0.25–0.5 x 10$^6$ cells/ml in flat bottomed 96 well plates (200 ml/ well; Sigma/Costar, St. Louis, MO).

## T regulatory cell assays

For in vitro suppression assays, CD4$^+$ CD25$^{high}$ Neuropilin$^+$ Treg cells were sorted from WT and one-allele *Stat5a-* or *Stat5b*-deficient mice. Naive, CD4$^+$ CD44$^{low}$ CD25$^-$ responder cells were sorted from congenic CD45.1 mice and labelled with Carboxyfluorescein succinimidyl ester (CFSE; Sigma-Aldrich). CD11c$^+$ antigen presenting cells (APCs) were purified from WT mice using positive selection beads (Miltenyi Biotec). 5 x 10$^4$ CD4$^+$ responder cells were stimulated with soluble anti-mouse CD3ε (1 μg/ml) in round bottom 96-well plates containing 1 x 10$^4$ APCs and varying numbers of Treg cells, ranging from 5 x 10$^4$ (1:1 ratio) to 1.56 x 10$^3$ (1:32 ratio). After 96 hr, cells were stained with fluorochrome-labelled anti-mouse CD4, CD45.1, and CD25. Percent suppression was calculated relative to WT controls and reflects the percentage of responder cells exhibiting at least one cell division. For 'Treg only' cultures, cells were stimulated with anti-CD3 and anti-CD28 in the presence human IL-2 (100 units/ml) for 72 hr. For iTreg differentiation, naive CD4$^+$ CD44$^{low}$ CD25$^-$ cells were sorted and cultured for 72 hr in the presence anti-CD3, anti-CD28, human TGF-β (10 ng/ml; R&D Systems, Minneapolis, MN), human IL-2 (100 units/ml) and anti-mouse IL-2, IL-4 and IFN-γ.

## RNA sequencing and transcriptome analysis

Cell sorting was used to purify cells from pooled lymph nodes and spleens of WT and one-allele *Stat5a-* or *Stat5b*-deficient mice (>99% purity). Ex vivo groups included naive T cells (CD4$^+$ CD44$^{low}$ CD25$^-$) and Treg cells (CD4$^+$ CD25$^{high}$ Neuropilin$^+$). In vitro groups included naive T cells that were treated with mouse IL-7 for 18 hr, effector T cells that were stimulated with anti-CD3 and anti-CD28 in the presence of human IL-2 for 72 hr, and induced Treg cells. All cultures included anti-mouse IL-2, IL-4 and IFN-γ (10 μg/ml each). Equal numbers of cells (0.5–2.5 x 10$^5$) were collected for each replicate. These were lysed in Trizol reagent and total RNA isolated by phenol-chloroform extraction with GlycoBlue as co-precipitant (7-15 μg per sample; Life Technologies). Single-end libraries were prepared with 0.1–0.5 μg of total RNA using the TruSeq RNA Sample Preparation Kit V2 and sequenced for 50 cycles with a HiSeq 2500 instrument (4–6 samples multiplexed per lane; Illumina, San Diego, CA). 50 bp reads were then mapped onto mouse genome build mm9 using TopHat and further processed using Cufflinks (*Garber et al., 2011*). 2–3 biological replicates were sequenced per genotype for every cell type and culture condition. QC-passing read counts are presented in *Supplementary file 1*.

Datasets are normalized based on RPKM (reads per kilobase exon model per million mapped reads) and purged of micro-RNAs, sno-RNAs and sca-RNAs. To minimize fold-change artifacts

caused by low abundance transcripts, a small offset (0.2–0.3; equal to the second quartile of each dataset) was added to all RPKM values (*Warden, Yuan, and Wu, 2013*). When multiple fragments were detected for a single gene, only the most abundant (i.e. highest average RPKM across all 3 genotypes) was considered for downstream analyses. Transcripts with RPKM values of less than 1 for all genotypes within a given cell type or condition were excluded. Fold change and variance across genotypes and biological replicates were calculated using EdgeR (*Robinson, McCarthy, and Smyth, 2009*). Transcripts were classified as differentially expressed if they exhibited a >1.5 fold change and significant pairwise variance (p<0.05) relative to WT controls. The 500 transcripts with greatest variance within each cell type or condition were used for multidimensional scaling (MDS) using the RobiNA software package (*Lohse et al., 2012*).

A 'paralog preference' scale was devised to illustrate the relative impact of *Stat5a* - or *Stat5b*-deficiency. First, all transcripts that were differentially expressed in *Stat5a*- or *Stat5b*- deficient cells (relative to WT controls) were pooled to generate a single list of STAT5-regulated genes for each cell type or condition. Next, the absolute fold change was calculated and multiplied by the higher of the two RPKMs (WT or KO), thereby generating a 'paralog score'. Note that the use of absolute fold change negates the distinction between up- and down-regulated genes, while the multiplication step improves the score for high-abundance transcripts. The paralog score for STAT5B was then divided by the paralog score for STAT5A and the resulting 'preference score' was log 2 transformed so that transcripts which are more impacted by the loss of STAT5B are assigned positive values while those which are more impacted by the loss of STAT5A are assigned negative values. All transcripts were then segregated into 12 bins according to preference scores (Bin 1 includes values of less than -5, Bin 2 ranges from -5 to -4, and so on). Data are displayed as histograms and the median preference score is indicated.

To identify 'paralog-specific' transcripts, we first identified those exhibiting >1.5 fold change and significant variance (p<0.05) when comparing *Stat5a*- or *Stat5b*-deficient cells directly to one another. Next, we refined this list by stipulating that transcripts must be differentially expressed in one KO relative to WT controls (>1.5 fold change) but not in the other (<1.2 fold change). Rare transcripts with opposite expression patterns (i.e. up-regulated in one genotype but down-regulated in the other) were excluded. Data are presented as pie charts.

All volcano plots, XY plots, histograms and pie charts were generated with the DataGraph software suite (Visual Data Tools, Inc.). Heat maps were generated with Multi Experiment Viewer (MeV; J. Craig Venter Institute, La Jolla, CA). Genome browser files (BigWig format) were processed to remove intronic reads using TopHat and are displayed with the Integrative Genomics Viewer (IGV; Broad Institute, Cambridge, MA).

GSEA analysis was performed as described (*Subramanian et al., 2005*). Unabridged RNA-seq datasets were used in conjunction with the following user-generated Gene Sets: 1) Treg signature genes (132 members)(*Hill et al., 2007*), 2) IL-2-regulated, STAT5A-dependent genes (258 members) (from the comparison of WT and 'one copy' *Stat5a*-deficient T cells; *Figure 6*), 3) IL-2-regulated, STAT5B-dependent genes (329 members)(from the comparison of WT and 'one copy' *Stat5b*-deficient T cells; *Figure 6*). Enrichment score curves and member ranks were generated by the GSEA software (Broad Institute). Normalized enrichment score (NES), false discovery rate (FDR) and nominal p Value is shown on each plot.

See *Supplementary file 2* for RPKM, fold change and p values for all experimental groups and conditions, *Supplementary file 3* for paralog preference calculations and *Supplementary file 4* for paralog-specific genes.

## Chromatin immuno-precipitation and DNA sequencing

Cell sorting was used to purify naive CD4$^+$ CD44$^{low}$ CD25$^-$ cells from WT, *Stat5a/b*$^{het}$ and two-allele *Stat5a*- or *Stat5b*-deficient mice (>99% purity). These were stimulated with anti-CD3 and anti-CD28 in the presence of human IL-2 for 48 hr (10 U/ml with anti-mouse IL-2, IL-4 and IFN-γ), then pulsed with IL-2 (100 U/ml) for one hour before fixing with 1% formaldehyde. They were then lysed (1 x 10$^7$ cells/sample), sonicated and immuno-precipated using a polyclonal rabbit anti-mouse IgG that recognizes both STAT5A and STAT5B (ab7969; Abcam, Cambridge, MA). Recovered STAT5-bound DNA fragments, along with un-precipitated 'input controls', were blunt-end ligated to adaptors and single-end libraries constructed using the NEBNext ChIP-Seq Library Prep for Illumina kit (New England Biolabs, Ipswich, MA). Sequencing was performed on a HiSeq 2500 instrument (50 cycles;

Illumina) and short reads (50 bps) aligned using Bowtie (*Langmead et al., 2009*). Non-redundant reads were mapped to the mouse genome (mm9) and aggregated into peaks and using MACS 1.4.2 (*Feng et al., 2012*). Only peaks with >3 fold enrichment over background and p values <0.00005 were called. Positive false discovery rates, or *q*-values, were calculated empirically for each peak and all were below 0.2% (*Storey, 2003*). 2 biological replicates were sequenced per genotype. 'bamCorrelate' from deepTools 1.5 was used to calculate Spearman's rank correlation coefficients as a measure of inter-replicate variability (WT=0.82, *Stat5a/b^het^*=0.83, *Stat5a*-deficient=0.82, *Stat5b*-deficient=0.81; all pairwise p-values <2.2. x 10^{-16})(*Ramirez et al., 2014*). Read depth for all replicates is presented in *Supplementary file 1*.

Peaks were annotated to the nearest known gene using HOMER (*Zhang et al., 2008*; *Heinz et al., 2010*). Localization was calculated as the percentage of peaks found within 10 kb intervals of the nearest transcriptional start sites and plotted as histograms. Direct comparison between experimental groups (i.e. peak overlap) was done with PAPST (*Bible et al., 2015*). Circos plot was generated by inputing the number of shared peaks between experimental groups to the Circos Table Viewer (http://mkweb.bcgsc.ca/tableviewer)(*Krzywinski et al., 2009*). Violin plot was generated by inputing tag numbers to the online BoxPlotR applet (http://boxplot.tyerslab.com)(*Editorial, 2014*). Transcription factor motif analysis was done with HOMER using an 'in house' database generated by applying de novo motif discovery to published ChIP-seq datasets. Genome browser files are displayed with IGV.

## ENCODE analysis

Strand-specific RNA sequencing data were generated by the ENCODE Transcriptome Group from Cold Spring Harbor Laboratories (U.S.A.) and the Center for Genomic Regulation (Spain)(https://genome.ucsc.edu/cgi-bin/hgTrackUi?hgsid=424400999_9OI4vJsT1sakRAPyi9mNSC7V81zc&g=wgEncodeCshlLongRnaSeq). DNAseI hyper-sensitivity data was generated by the University of Washington ENCODE group (https://genome.ucsc.edu/cgi-bin/hgTrackUi?hgsid=424401115_qgaAWZ6Xs38F1laFE3UuHnvAG7AS&g=wgEncodeUwDnase). Data are used in accordance with the ENCODE data release policy (*Yue et al., 2014*) and visualized with the UCSC genome browser, focusing on the mouse *Stat5a/b* locus (chr11:100642045-100746483).

## Retroviral gene transduction

Retroviral vector expressing phosphatase-insensitive STAT5A was generated as described (*Zhu et al., 2003*). Plasmids were transfected into Phoenix packaging cells using Lipofectamine (Invitrogen) and the resulting viral supernatants used to transduce CD4^+ cells from WT, *Stat5a/b^het^* or one-allele *Stat5b*-deficient mice. These were stimulated (anti-CD3/CD28) in the presence of anti-mouse IL-2 for 48 hr, exposed to viral supernatant for 1 hr (at 2200 rpm, 18°C), and cultured for an additional 48 hr in the presence of human IL-2 (100 U/ml). For some experiments, cells were cultured under iTreg (10 ng/ml human TGF-β) or Th17 (2.5 ng/ml human TGF-β + 20 ng/ml mouse IL-6) polarizing conditions before and after transduction (both in the presence of anti-mouse IL-4 and IFN-γ). For RNA-seq, 1-2 x 10^5 cells expressing the bicistronic transduction marker (human NGFR) were purified by cell sorting. Transcripts that were significantly impacted (>1.5 fold change, p<0.05) by ectopic STAT5A relative to empty vector were enumerated using EdgeR.

## Relative paralog measurements

Transcriptome data for CD4^+ naive and Treg cells was sourced from: 1) Immunological Genome Project (mouse microarrays: http://www.immgen.org), 2) EMBL-EBI Expression Atlas (mouse RNA-seq: https://www.ebi.ac.uk/gxa/experiments/E-MTAB-2582), 3) BioGPS Primary Cell Atlas (human microarrays: http://biogps.org/dataset/BDS_00013/), and 4) our RNA-seq catalogue (described above; *Figure 6*). Normalized expression values (microarray signal intensity or FPKM) for *Stat5a* and *Stat5b* were first divided by one another to generate a paralog ratio which was then converted to a percentage (% total STAT5 mRNA accounted for by each paralog) and presented as pie charts.

Total STAT5 protein was measured by flow cytometry in naive (CD3ε^+ CD4^+ CD44^low IL-7R^+), Treg (CD3ε ^+ CD4^+ FOXP3^+), Tfh (CD3ε ^+ CD4^+ PD1^+ CXCR5^high) and effector/memory (CD3ε ^+CD4^+CD44^high) T cells from one- or three-allele *Stat5a*- or *Stat5b*-deficient mice, as well as *Stat5a/b^het^* mice and WT controls. Mean fluorescence intensity (MFI) was first divided by the baseline (i.e.

WT controls) to generate 'fold change' values which, in turn, were divided across *Stat5a*- and *Stat5b*-deficient genotypes bearing the same total number of alleles. The resulting paralog ratios (one for one-allele cells and one for three-allele cells) were then averaged, converted to a percentage (% total STAT5 protein accounted for by each paralog) and presented as pie charts.

## Statistics

Unpaired ANOVA was used to quantify statistical deviation between experimental groups. In all figures, black asterisks denote significant differences ($p<0.05$) between the indicated group and WT controls. Orange asterisks denote significant differences between *Stat5a*- and *Stat5b*-deficient mice bearing the same total number of STAT5 alleles.

## Data deposition

All sequencing data have been deposited to the Gene Expression Omnibus under the accession number GSE77656.

## Acknowledgements

We thank members of the O'Shea lab for helpful discussions, Victoria Hoffman for pathology scoring, Gustavo Gutierrez-Cruz for sequencing and the NIAMS Flow Cytometry Group for cell sorting. This study utilized the high-performance computational capabilities of the Helix System at the National Institutes of Health, Bethesda, MD (http://helix.nih.gov) and was supported by NIDDK and NIAMS intramural research programs. JJO'S and the NIH hold patents related to therapeutic targeting of Jak kinases and have a Collaborative Research Agreement and Development Award with Pfizer Inc.

## Additional information

### Competing interests

JJO: Holds NIH patents related to therapeutic targeting of Jak kinases and have a Collaborative Research Agreement and Development Award with Pfizer Inc. The other authors declare that no competing interests exist.

### Funding

| Funder | Author |
| --- | --- |
| National Institutes of Health | Alejandro Villarino |
| | Arian Laurence |
| | Gertraud W Robinson |
| | Michael Bonelli |
| | Barbara Dema |
| | Behdad Afzali |
| | Han-Yu Shih |
| | Hong-Wei Sun |
| | Stephen R Brooks |
| | Lothar Hennighausen |
| | Yuka Kanno |
| | John J O'Shea |

The funders had no role in study design, data collection and interpretation, or the decision to submit the work for publication.

### Author contributions

AV, Conception and design, Acquisition of data, Analysis and interpretation of data, Drafting or revising the article; AL, BA, Acquisition of data, Analysis and interpretation of data, Drafting or revising the article; GWR, LH, Conception and design, Drafting or revising the article; MB, BD, H-YS, H-WS, SRB, Acquisition of data, Analysis and interpretation of data; YK, Analysis and interpretation of data, Drafting or revising the article; JJO, Conception and design, Analysis and interpretation of data, Drafting or revising the article

**Author ORCIDs**

Alejandro Villarino, http://orcid.org/0000-0001-8068-2176

**Ethics**

Animal experimentation: Animals were handled in accordance with NIH guidelines and all experiments approved by the NIAMS Animal Care and Use Committee (Animal Study Number: A013-10-07).

# Additional files

## Supplementary files

• Supplementary file 1. *Read depth for sequencing experiments.* Spreadsheet reports QC-passing reads for all RNA-seq and ChIP-seq experiments.

• Supplementary file 2. *RPKM values for transcriptome analysis of STAT5-deficient CD4+ T cells.* Spreadsheet includes RPKM values for all experimental groups and biological replicates.

• Supplementary file 3. *Paralog preference calculations for STAT5-deficient CD4+ T cells.* Spreadsheet includes paralog preference calculations for all relevant cell types and experimental conditions.

• Supplementary file 4. *STAT5 paralog-specific genes.* Spreadsheet lists STAT5 paralog-specific genes for all relevant cell types and experimental conditions.

## Major datasets

The following dataset was generated:

| Author(s) | Year | Dataset title | Dataset URL | Database, license, and accessibility information |
|---|---|---|---|---|
| Villarino AV, Sun HW, Gutierrez-Cruz G, Kanno Y, O'Shea JJ | 2016 | STAT5 paralog dose governs T cell effector and regulatory function | http://www.ncbi.nlm.nih.gov/geo/query/acc.cgi?acc=GSE77656 | Publicly available at the NCBI Gene Expression Omnibus (accession no: GSE77656). |

The following previously published datasets were used:

| Author(s) | Year | Dataset title | Dataset URL | Database, license, and accessibility information |
|---|---|---|---|---|
| Mouse ENCODE Consotium | 2015 | Mouse ENCODE RNAseq Data | http://www.ncbi.nlm.nih.gov/geo/query/acc.cgi?acc=GSE39524 | Publicly available at the NCBI Gene Expression Omnibus (Accession no: GSE39524) |
| Mouse ENCODE Consotium | 2015 | Mouse ENCODE DHS Data | http://www.ncbi.nlm.nih.gov/geo/query/acc.cgi?acc=GSE40869 | Publicly available at the NCBI Gene Expression Omnibus (Accession no: GSE40869) |

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
