## [Decision Letter]

Thank you for submitting your work entitled "STAT5 paralog dose governs T cell effector and regulatory function" for peer review at *eLife*. Your submission has been favorably evaluated by Tadatsugu Taniguchi (Senior editor), and three reviewers, one of whom is a member of our Board of Reviewing Editors.

The reviewers have discussed the reviews with one another and the Reviewing editor has drafted this decision to help you prepare a revised submission.

Summary:

Villarino et al. address specific roles of the STAT5 paralogs on regulatory and helper/effector T cell function, particularly with respect to production of members of cytokine family that signals through receptors bearing common γ chain. Largely through genomic analysis of a range of allelic combinations of *Stat5a* and *Stat5b* mutant mice via disruption of either single or dual alleles (combinations of homozygous or heterozygous *Stat5a* and *Stat5b* mutant mice), the authors find that the two paralogs are highly redundant in their functions, but because STAT5B is expressed at higher levels, it has a dominant role in regulating immune cell development and function. The studies are generally well-performed and address an issue that has not been clearly resolved for immune cells.

Essential revisions:

1) Although the authors have used state-of-art genomic analysis to arrive at the conclusion that a dose-dependent effect of the STAT5 paralog is critical to drive the STAT5 inducible genes optimally, most of their conclusions are correlative rather serving as a direct biological 'cause-effect' relation. Additional studies would be needed to provide direct biological evidence to validate their claim of dosage effect of paralog on T cell-mediated immune response. The studies shown in Figure 9 should be extended to demonstrate that the aberrant T cell responses exhibited by STAT5B-deficieny (e.g., increased frequency of CD44high effector populations, lower Tbet induction in response to IL-2, lower extent of IL-2Ra and FoxP3 induction during Treg development and greater loss of suppressive ability of Tregs) can be reversed by retroviral expression of STAT5A in STAT5B-deficient cells. This could be achieved by transducing naive T cells ex vivo and then analyzing their ability to differentiate into distinct T cell lineages.

2) This paper involves comparisons between 8 different mouse genotypes yet according to the Methods section these were all compared using single t-tests which is inappropriate. The analysis of data in Figure 1, Figure 2, Figure 2—figure supplement 1, Figure 3, Figure 4, Figure 4—figure supplement 1, Figure 5, Figure 6, and Figure 9 should be performed using ANOVA or some other comparable test to account for multiple sample comparison.

3) Replicates for ChIP-seq experiments should be performed if not done so already. Without replicates a determination of FDR (or IDR using Encode methods for combining replicates) cannot be performed. This is important given that the question is whether or not STAT5A and STAT5B bind differentially to any regions of the genome. A Table summarizing read depth is needed to allow assessment of data quality. An anonymous link to a browser session is recommended. The RNA-Seq and ChIP-seq data needs to be deposited in a public repository.

4) Because of the global nature of the *Stat5* knockouts, cell types in addition to T cells could contribute autoimmune kidney pathology, particularly macrophages. These experiments would be strengthened by blocking cytokine signaling pathways of T cells that are dysregulated in the absence of STAT5B. At the least, the potential contributions of additional cell types to the renal pathology needs to be discussed.

5) A major point of the manuscript is that STAT5B is dominant because it is expressed at higher levels. This point is not adequately documented. The only figure panel that directly shows differential expression at the level of protein is Figure 9, which uses a pan-STAT Ab to visualize STAT5 in WT and 5A or 5B KO cells. The labeling of this figure is ambiguous, but a large decrease in expression is observed for 5A (5A-deficient) cells and no expression is seen in 5B (5B-deficient) cells. These differences seem to be much greater than the ~2-fold differences in RNA expression shown in Figure 9 for Treg and Nv cells and RNA expression is nearly equal in Tef cells. The authors show evidence for cell-specific enhancers in Figure 9—figure supplement 2, which could explain differences in STAT5 expression across cell types, but these do not necessarily explain differences in STAT5A vs. STAT5B expression within a particular cell type, raising the possibility that differences in post transcriptional regulation of STAT5A/B expression are important. Further documentation of different levels of STAT5B and STAT5A protein expression is needed in wild type cells. Any further insights into the basis of differential expression would also strengthen the manuscript.

---

## [Author Response]

Essential revisions:

*1) Although the authors have used state-of-art genomic analysis to arrive at the conclusion that a dose-dependent effect of the STAT5 paralog is critical to drive the STAT5 inducible genes optimally, most of their conclusions are correlative rather serving as a direct biological 'cause-effect' relation. Additional studies would be needed to provide direct biological evidence to validate their claim of dosage effect of paralog on T cell-mediated immune response. The studies shown in Figure 9 should be extended to demonstrate that the aberrant T cell responses exhibited by STAT5B-deficieny (e.g., increased frequency of CD44high effector populations, lower Tbet induction in response to IL-2, lower extent of IL-2Ra and FoxP3 induction during Treg development and greater loss of suppressive ability of Tregs) can be reversed by retroviral expression of STAT5A in STAT5B-deficient cells. This could be achieved by transducing naive T cells ex vivo and then analyzing their ability to differentiate into distinct T cell lineages.*

A principal finding of our work is that STAT5B is dominant over STAT5A because it is more abundant, not because it is endowed with unique functional properties. This conclusion is supported in the original manuscript by gain-of-function studies showing that ectopic STAT5A can rescue gene expression in *Stat5b-*deficient T cells but, as noted by the reviewers, these did not address whether it can also rescue cellular differentiation. To explore the latter possibility, we transduced *Stat5b*-deficient T cells under Treg or Th17 polarizing conditions, then measured FoxP3 and IL-17 to gauge lineage commitment. As expected, ectopic STAT5A restored their capacity to become FoxP3^+^ Treg cells while, at the same time, limiting their capacity to become IL-17-producing effectors. Ectopic STAT5A even drove FoxP3 expression in WT and heterozygous cells (i.e. *Stat5a*^+/-^*Stat5b*^+/-^), suggesting that availability of STAT5 is actually rate limiting for Treg differentiation. Furthermore, there was a clear, linear correlation between STAT5 protein levels, whether endogenous or ectopic, and FoxP3 induction in all genotypes, suggesting that high concentrations predispose towards the Treg lineage. Collectively, these data support our hypothesis that STAT5 paralog dose bears major influence on T cell differentiation and imply that a certain STAT5 threshold must be achieved to execute the Treg program.

All ectopic STAT5A experiments have been compiled in the new Figure 10 and Figure 10—figure supplement 1. They are described in the Results section (last two paragraphs).

*2) This paper involves comparisons between 8 different mouse genotypes yet according to the Methods section these were all compared using single t-tests which is inappropriate. The analysis of data in Figure 1, Figure 2, Figure 2—figure supplement 1, Figure 3, Figure 4, Figure 4—figure supplement 1, Figure 5, Figure 6, and Figure 9 should be performed using ANOVA or some other comparable test to account for multiple sample comparison.*

Unpaired ANOVA is now used for all analyses involving three or more experimental groups. This methodology clearly improved the statistical power of our multi-group comparisons – in some cases revealing differences that were missed by single t-tests – but did not affect our interpretation of the results.

*3) Replicates for ChIP-seq experiments should be performed if not done so already. Without replicates a determination of FDR (or IDR using Encode methods for combining replicates) cannot be performed. This is important given that the question is whether or not STAT5A and STAT5B bind differentially to any regions of the genome. A Table summarizing read depth is needed to allow assessment of data quality. An anonymous link to a browser session is recommended. The RNA-Seq and ChIP-seq data needs to be deposited in a public repository.*

Two independent ChIP-seq experiments have been performed and Spearman's rank coefficients indicate good correlation between biological replicates (WT=0.82, *Stat5a/b*^het^=0.83, *Stat5a*-deficient=0.82, *Stat5b*-deficient=0.81). Positive false discovery rates, or q-values, were calculated empirically for each peak. Due to stringent peak calling criteria – only peaks with p values <0.00005 were considered – all were more than 3-fold enriched over background with FDR values below 0.2%. Each of these points are now clarified in the Methods section (subsection “Chromatin immuno-precipitation and DNA sequencing”).

QC-passing read counts for all RNA-seq and ChIP-seq replicates are now provided in [Supplementary-material SD1-data].

Although a dedicated UCSC genome browser session would facilitate access to our data, we are currently unable to allocate the server space necessary to host it in perpetuity. However, the raw data have been deposited to the Gene Expression Omnibus and, thus, will be freely available upon publication (accession number GSE77656; http://www.ncbi.nlm.nih.gov/geo/query/acc.cgi?token=mvyzwwievhcbvgr&acc=GSE77656).

*4) Because of the global nature of the Stat5 knockouts, cell types in addition to T cells could contribute autoimmune kidney pathology, particularly macrophages. These experiments would be strengthened by blocking cytokine signaling pathways of T cells that are dysregulated in the absence of STAT5B. At the least, the potential contributions of additional cell types to the renal pathology needs to be discussed.*

Given the importance of helper T cells for maintaining and/or breaking immunological tolerance, and extensive data (including our own) on T cell intrinsic functions of STAT5, we conclude that helper T cell are the principal instigators of autoimmunity in *Stat5b*-deficient mice. However, as noted by the reviewers, STAT5 is known to impact many other cell types that may influence pathogenesis in these germline mutant animals. This important point is now raised in the Discussion section:

"These findings strongly implicate Tfh cells in the pathogenesis of Stat5b-deficient mice but, since these are germline ‘knockouts’, we must also consider the (likely) possibility that intrinsic defects in other cell types contribute to their autoimmune phenotype. For instance, multiple dendritic cell subsets are known to be dysregulated in mice lacking STAT5 or upstream kinase Jak3, and its influence on non-immune cells, particularly downstream of hormone receptors, cannot be overlooked."

5) A major point of the manuscript is that STAT5B is dominant because it is expressed at higher levels. This point is not adequately documented. The only figure panel that directly shows differential expression at the level of protein is Figure 9, which uses a pan-STAT Ab to visualize STAT5 in WT and 5A or 5B KO cells. The labeling of this figure is ambiguous, but a large decrease in expression is observed for 5A (5A-deficient) cells and no expression is seen in 5B (5B-deficient) cells. These differences seem to be much greater than the ~2-fold differences in RNA expression shown in Figure 9 for Treg and Nv cells and RNA expression is nearly equal in Tef cells. The authors show evidence for cell-specific enhancers in Figure 9—figure supplement 2, which could explain differences in STAT5 expression across cell types, but these do not necessarily explain differences in STAT5A vs. STAT5B expression within a particular cell type, raising the possibility that differences in post transcriptional regulation of STAT5A/B expression are important. Further documentation of different levels of STAT5B and STAT5A protein expression is needed in wild type cells. Any further insights into the basis of differential expression would also strengthen the manuscript.

The idea that STAT5B is more abundant than STAT5A is central to our thesis on why it is the dominant paralog in helper T cells. To better support this point, we employed two independent strategies. First, we mined published transcriptome datasets to confirm that STAT5B is more abundant at the mRNA level (Figure 9—figure supplement 1). Second, we refined our pan-STAT5 flow cytometry assay to better illustrate the disparity at the protein level (Figure 9).

Due to inevitable differences in affinity and/or avidity, we believe it is impossible to compare cellular concentrations of two proteins using two separate antibodies. Therefore, we have chosen to calculate relative abundance based on the performance of a single pan-STAT5 antibody, which presumably recognizes a single homologous epitope, across a range of STAT5 allele combinations. Using this methodology, which has been refined such that we can resolve individual subsets (i.e. naive, Treg, Tfh) across 6 genotypes (or more), we confirmed that STAT5B makes the greater contribution to the total STAT5 pool. The asymmetry was most obvious in cells lacking one allele – deleting a single copy of *Stat5b* was enough to yield a phenotype while deleting a single copy *Stat5a* was not – but was also evident in those lacking two; *Stat5b*-deficient cells had far less STAT5 protein than *Stat5a*-deficient cells (Figure 9). We also found that STAT5 levels were consistently lower in cells from 'one allele' Stat5a-deficient mice (*Stat5a*^-/-^*Stat5b*^+/-^) than in those from double heterozygotes (*Stat5a*^+/-^*Stat5b*^+/-^), suggesting that, while *Stat5b* may be the dominant input, the *Stat5a* locus does make an appreciable, albeit more limited, contribution (Figure 9).

All STAT5 mRNA and protein measurements have been compiled in the new Figure 9 and Figure 9—figure supplement 1. They are described in the Results section, subsection “Asymmetric expression of STAT5 paralogs in helper T cells” and in the Methods section, subsection “Flow cytometry”.

Overall, there was good accord between our mRNA and protein data in demonstrating that STAT5B is from 1.5 to 3 times more abundant than STAT5A. These ratios are in line with our original flow cytometry studies (Original Figure 9) but less dramatic than what we had observed by western blotting (Original Figure 9), likely due to differences in sensitivity between the two assays. Given this inconsistency, and the fact that our original flow cytometry studies did not distinguish individual T cell subsets, we have chosen to include only our latest flow cytometry data in the revised manuscript.

One surprise that emerged from our RNA-seq studies is that paralog asymmetry appeared to be lost after in vitro activation (see 'effector T cells' in the Original Figure 9). Upon inspection, we found that this was evident only when cells were cultured for 72 hours, not when they were cultured for 96 hours (Data not shown) or under iTreg polarizing conditions (data from Figure 5). We also found no evidence for such an effect in published datasets (Data not shown), nor could we detect it at the protein level (Data not shown). Thus, while it remains possible that acute stimuli (i.e. TCR engagement, co-stimulation) temporarily alter the balance between STAT5 paralogs, we are unsure about the kinetics and/or robustness of this phenomenon and, thus, have decided to omit all mention of it from the revised manuscript.